# Effects of Coffee on the Gastro-Intestinal Tract: A Narrative Review and Literature Update

**DOI:** 10.3390/nu14020399

**Published:** 2022-01-17

**Authors:** Astrid Nehlig

**Affiliations:** 1INSERM U 1129, Pediatric Neurology, Necker-Enfants Malades Hospital, University of Paris Descartes, 75015 Paris, France; nehliga@unistra.fr; 2Faculty of Medicine, INSERM U 1129, 67000 Strasbourg, France

**Keywords:** coffee, gastro-intestinal tract, gastro-esophageal reflux, gallstones, colon motility, microbiota, cancer

## Abstract

The objective of the present research was to review the state of the art on the consequences of drinking coffee at the different levels of the gastrointestinal tract. At some steps of the digestive process, the effects of coffee consumption seem rather clear. This is the case for the stimulation of gastric acid secretion, the stimulation of biliary and pancreatic secretion, the reduction of gallstone risk, the stimulation of colic motility, and changes in the composition of gut microbiota. Other aspects are still controversial, such as the possibility for coffee to affect gastro-esophageal reflux, peptic ulcers, and intestinal inflammatory diseases. This review also includes a brief summary on the lack of association between coffee consumption and cancer of the different digestive organs, and points to the powerful protective effect of coffee against the risk of hepatocellular carcinoma. This review reports the available evidence on different topics and identifies the areas that would most benefit from additional studies.

## 1. Introduction

In this review, we will consider the effects of coffee ingestion on the organs composing the gastro-intestinal tract, which are the first organs with which coffee and its large diversity of components come into contact after ingestion. The influence of coffee on digestive processes has been known for a long time, and drinking coffee after a meal has become a habit for most of us. Indeed, coffee is considered to favor digestion by acting on the acid production of the stomach, on bile and pancreatic secretion, and on colon motility. Here, we will develop the consequences of coffee intake at the different steps of and on the different organs involved in digestion, mainly in healthy individuals but also in those afflicted with some specific gastrointestinal conditions. This review is not intended to consider in detail all the mechanisms of action on coffee on these processes, since these aspects have been developed mainly in two detailed and recent reviews [1,2]. Furthermore, the knowledge on these aspects remains limited mainly because concentrations of the different components of coffee are strongly affected by various factors such as the coffee type and origin, the roasting process and the method of preparation, which largely vary among different countries. These differences generate marked difficulties in the attribution of the effect of coffee to one specific compound and in their extension to the population worldwide.

## 2. Coffee and Enzymatic Salivary Secretion

Coffee consumption activates the secretion of salivary alpha-amylase (sAA), an enzyme involved in polysaccharide digestion. sAA measurement is currently used also as an index for soothing/relaxation [3] and as a marker of the stress response [4]. In healthy young individuals, the administration of caffeine was reported to either activate [5] or not influence [6] the secretion of sAA.

In a recent randomized, double blind, crossover clinical trial on 40 healthy individuals, 20 women and 20 men, 4 different coffee types (200 mL containing 160 mg caffeine in warm filtered coffee, warm and cold instant coffee, cold espresso administered 1 week apart) increased sAA activity and salivary gastrin release, showing a peak at 30 min post-ingestion and remaining stable up to 60 min. The largest increase in sAA was seen with cold instant coffee, closely followed by cold espresso. Slightly lower sAA levels were found at 30 min post-ingestion, with warm instant coffee followed by filtered coffee. These data indicate a possible effect of coffee temperature on sAA activity, but this effect has not been studied elsewhere. All these effects were independent of gender and had no effect on self-reported gastro-intestinal symptoms. Changes in sAA in gastrin secretion occurred without affecting cortisol levels, indicating that coffee ingestion did not increase stress levels [7].

## 3. Coffee Stimulates Gastric Secretion but Does Not Accelerate Gastric Emptying

When food arrives in the stomach, gastric glands secrete hydrochloric acid and enzymes, such as pepsine, chymosine, and lipase, that initiate the digestion of carbohydrates, proteins, and lipids. Coffee (via caffeine and other components, mainly polyphenols) stimulates the secretion and production of gastrin and hydrochloric acid. Caffeine stimulates gastrin secretion in preparations of pure rat G cells (gastrin cells) that are present in the stomach and duodenum and secrete gastrin [8]. Caffeinated coffee, especially ground coffee but also instant caffeinated coffee, stimulates more effectively gastrin secretion than decaffeinated coffee, pointing to the role of caffeine [9,10]. However, older studies in humans in vivo reported either a lack of stimulation by caffeine [11] or similar magnitude of stimulation by coffee and decaffeinated coffee which was larger than that with caffeine [12], reflecting the action of both caffeine and other components of coffee on gastrin secretion. Others reported that the stimulatory properties of coffee on gastrin secretion appear to be partially lost during decaffeination [11,13]. Adenosine was shown to inhibit gastric acid secretion. In the rat stomach, this inhibitory effect may be mediated by indirect increase in the release of somatostatin, consequently controlling gastric acid secretion [14].

Gastric acid secretion is reduced in human gastric cancer cells by exposure to coffee-brew representative concentrations of N-methylpyridinium, which impairs the expression of prosecretory gastric acid secretion [15]. This effect depends on the degree of roasting, with dark roasted coffee being less effective in stimulating gastric acid release, possibly because of the presence of a higher amount of N-methylpyridinium and smaller amounts of chlorogenic acids, trigonelline, and ^β^N-alkanoyl-5-hydroxytryptamides (C5HTs). It is not clear yet whether a high ratio of N-methylpyridinium to these other components represents a critical factor in the reduction of coffee-associated gastric acid secretion [16,17].

Coffee does not influence the rapidity of stomach emptying [18,19,20,21,22,23]. Only one study reported that the half-time of gastric emptying was reduced to 154 min in subjects receiving caffeine in comparison with 182 min for the control condition [24]. There are no available data on other coffee components.

## 4. Risk of Gastro-Esophageal Pathology

The action of coffee on stomach acid secretion has raised the issue of a possible increase in dyspepsia (poor digestion, discomfort, nausea, heartburns, eructation, and flatulence), or esophageal burns, gastritis or ulcers, and gastro-esophageal reflux disease (GERD). Diet plays an important role in heartburn, and many foods can relax the lower esophageal sphincter (LES) allowing food to escape into the esophagus and cause heartburn. Alterations of the structure and function of the LES may predispose to GERD [25].

### 4.1. Functional Dyspepsia

About 25% of the population suffers from functional dyspepsia (or non-ulcer stomach pain or non-ulcer dyspepsia) worldwide. Functional dyspepsia applies to recurring signs and symptoms of indigestion that have no obvious cause. This syndrome is often associated with the type of foods consumed. However, the exact causes of dyspepsia remain unclear. Patients with dyspepsia are advised to avoid aspirin and non-steroidal anti-inflammatory drugs (NSAIDs), smoking, and drinking alcohol and coffee, but it remains controversial whether these factors, and mainly coffee, may causally relate to the disease [26,27,28,29]. Several studies from Europe, America, and Australia have not found any relation between coffee consumption and dyspepsia [18,26,30], while others from the USA, Iran, and China found an association between coffee consumption and functional dyspepsia [31,32,33]. In a study by DiBaise [31], two types of coffee roasting were applied; one sample had undergone flash roasting lasting about 10 min and the second was subjected to conventional conduction roasting lasting 3–4 h. The potential impact of the two roasting processes on coffee composition was not studied. There was no difference in the effects of coffee on dyspepsia symptoms related to the type of roasting either after fasting or after a standard meal [31]. A recent review of 16 studies reported an association between caffeine and functional dyspepsia only in 4/16 studies, but coffee was not studied in this review [34]. In a single very recent study, the replacement of coffee by a non-caffeinated substitute was reported to improve the symptoms of functional dyspepsia in 51 patients suffering from this syndrome, but the components of coffee responsible for the effect on functional dyspepsia were not studied [35]. Often, smoking, taking aspirin or NSAIDs, eating spicy food, and infection by *Helicobacter pylori* seem to represent the major risk factors [19,28,29]. At this point, it is impossible to firmly conclude on a potential relationship between coffee and functional dyspepsia. This point warrants further investigation.

Conclusions about the relationship between coffee/tea consumption and the occurrence of gastro-intestinal symptoms are limited because these drinks are often consumed with or after a meal. Some authors even concluded that, due to contradictory data and the unclear relationship between the occurrence of gastro-intestinal symptoms and coffee consumption, it should not be routinely recommended to avoid coffee consumption in patients with gastro-intestinal symptoms [36].

### 4.2. Gastro-Esophageal Reflux (GERD)

GERD is an unpleasant reflux condition caused by the return of the acid stomach content into the esophagus. Coffee has been hypothesized to diminish basal lower esophageal sphincter (LES) pressure, which would lead to gastroesophageal reflux and heartburn, but this is not reported by all studies [19]. The most commonly reported cause of GERD is obesity, especially in women. 

The studies on LES pressure concerned limited groups of subjects, i.e., 31 healthy subjects [37]; 12 healthy subjects [38]; 20 normal volunteers and 16 patients [39]. In the latter three studies, as in another one [40], the consumption of regular coffee was associated with LES pressure decrease, symptoms of GERD, and heartburn. They are also described with tea [38,41,42,43,44], but not with decaffeinated coffee [36,39] and rarely with caffeine [45], suggesting that other components in coffee could be responsible for these effects. 

The effects of various types of coffee (ground coffee containing caffeine, decaffeinated coffee using ethyl acetate or methylene chloride, instant coffee processed at 45 °C or 150 °C, or steamed) on LES pressure, acid secretion, and blood gastrin were measured in eight subjects. A sustained decrease in LES was found, whether or not the coffee contained caffeine. Caffeinated ground coffee stimulated acid secretion more than decaffeinated ground coffee, but not more than a steam-treated caffeinated coffee whose acidity was reduced and irritant components were removed. The stimulation of acid secretion ability did not differ with the type of instant coffee. Ground coffee containing caffeine resulted in higher blood gastrin levels than other ground coffees. This study highlights that the variability in acid production in response to coffee consumption might partly originate in the type of processing of green coffee beans [10]. It must also be noted that GERD is linked to many different types of foods, such as mainly spicy and high-fat foods, beer, wine and alcohol, high-salt diet, carbonated soft drinks or beverages, citrus, coffee, and chocolate. A positive association has also been reported between GERD and increased body mass index [40], and weight loss is advised to reduce symptoms [46].

As detailed in Table 1, two meta-analyses [47,48] and 28 single studies looked at the effect of coffee on GERD. The outcome of the single studies was variable. Twelve studies were performed in Europe, five in America, ten in Asia, and one in Oceania. The data from the meta-analysis by Kim et al. [47] included 15 case-control studies and no association was found between coffee consumption and GERD. The authors used odds ratio (OR) for their analysis, which is a statistical value that quantifies the strength of the association between two events. They found an OR of 1.06 with a 95% confidence interval (95% CI) of 0.94–1.19). The 95% CI is used to estimate the precision of the OR, with lower values correlating with the precision of the OR. The OR of the low-intake group (<4 cups/day) reached 0.91 (95% CI = 0.82–1.01), while that of the high-intake group (>5 cups/day) was 1.14 (95% CI = 0.69–1.88). The ORs are not statistically significant between the groups with or without exposure to coffee. The second meta-analysis by Chen et al. [48] included 21 studies, and it also reported no association between coffee consumption and GERD symptoms when all studies were taken into consideration (Relative Ratio (RR) = 1.07; 95% IC = 0.96–1.19, *p* < 0.001). A lack of association was found in retrospective, prospective, Asian, questionnaire-based, and high-quality studies, while the risk of GERD was increased by coffee in cross-sectional studies where the RR was 1.54 (95% CI = 0.40–5.93; *p* < 0.001), with obviously a large heterogeneity between studies.

Fifteen single studies did not show any association between coffee consumption and GERD symptoms. This was true for the 14 case-control studies [42,44,49,50,51,52,53,54,55,56,57,58,59,60] and one single prospective study [61].

A decreased risk of GERD in coffee consumers was reported in two European studies whose outcomes varied with the group considered. The first found a significant reduction in GERD symptoms and their frequency not to be associated with coffee, whereas the same study reported an aggravation of GERD symptom frequency and severity associated with regular coffee consumption only in patients with severe or long-lasting GERD [62]. The second study found a dose-dependent reduced risk of GERD associated with coffee consumption, but only in male subjects, with no risk change in women [53].

In contrast, one review and meta-analysis of 40 studies [48] and 11 single studies reported an association between coffee consumption and GERD [35,41,43,62,63,64,65,66,67,68,69]. Most studies found a modest aggravation (10–20%), except for two Asian studies that reported 30–40% aggravation of GERD with coffee [43,66]. One study from the USA reported a 23% aggravation of GERD with at least six cups of caffeinated coffee and even a 48% aggravation of GERD symptoms with at least six cups of decaffeinated coffee. When two servings of coffee were replaced with water, GERD symptoms were normalized [68]. Finally, another US study reported that replacement of caffeinated or decaffeinated coffee by non-caffeinated coffee-substitute drink reduced four-fold the risk of GERD [35]. The composition of the substitute is unknown; it is only clear that caffeine was missing, which makes it difficult to identify which coffee component might be responsible for the potential effect of coffee on GERD.

A recent Taiwanese study gathered 1837 participants (970 men; age 51.6 ± 10.2 years) with data on clinical characteristics and consumption of coffee and tea with or without additives such as milk or sugar. Coffee or tea drinking was considered only if occurring at least 4 days/week for 3 months. Heavy coffee or tea consumption corresponded to drinking at least two cups/day. Among the 1837 subjects, 467 (25.4%) were diagnosed as having symptomatic GERD and 427 (23.2%) had erosive reflux disease (ERD) on endoscopic examination. Drinking coffee or tea was not associated with reflux symptoms or ERD, and this lack of association remained when coffee was drunk with milk or sugar [61].

Several studies have looked at the lifestyle factors related to GERD. A US study on 317 GERD patients and 182 asymptomatic controls found that risk factors for GERD were the consumption of soft drinks and tea [56]. One multi-center Indian study looked at 3224 subjects and reported that 245 persons (7.6%) suffered from GERD. These symptoms were related to older age, as well as consumption of non-vegetarian and fried food, carbonated drinks, and tea/coffee [44]. Another Indian study on 358 undergraduate students (mean age 20.3 years) found 115 participants with at least one episode of heartburn per week, but a GERD diagnosis was only made in 18 subjects; frequent consumption of carbonated drinks and of tea and coffee was significantly associated with a diagnosis of GERD [43]. On the other hand, a Chinese study on 2044 patients (aged 16–82 years) did not find any contribution of coffee and tea consumption to the development of esophagitis [42]. Finally, a Swedish study on 3153 individuals with GERD compared to 40,210 control subjects reported moderately strong inverse associations between the risk of reflux and exposure to coffee (possibly related to GERD), bread high in dietary fiber content, and frequent physical exercise, which are supposed to limit GERD symptoms, whereas the intake of alcohol (possibly causing GERD) or tea did not affect the risk of reflux [50].

Two studies looked at the risk of developing Barrett’s esophagus (BE) linked to coffee consumption. BE is an alteration of the mucosal cells of the esophagus lining due to frequent GERD and acidity. It may ultimately lead to esophagus cancer. A US study found no association between coffee consumption and the risk of BE when population was adjusted for confounders (sex and race), which led to an OR of 1.04 (95% IC = 0.76–1.42) [70]. A recent study examined 339 patients with BE, 462 patients with erosive esophagitis, and 619 controls. In this Italian population, BE risk was higher in former coffee drinkers, irrespective of levels of exposure (cups per day ≤ 1: OR = 3.76, 95% CI = 1.33–10.6; >1: OR = 3.79, 95% CI = 1.31–11.0) and was higher with duration (>30 years: OR = 4.18, 95% CI = 1.43–12.3) and for late quitters (≤3 years from cessation: OR = 5.95, 95% CI 2.19–16.2). The risk of BE was also higher in subjects who started drinking coffee later (age > 8 years: OR = 6.10, 95% CI = 2.15–17.3). No association was found in current drinkers, but the risk of erosive esophagitis was increased in light drinkers (<1 cup per day OR = 1.85, 95% CI = 1.00–3.43) [59].

Thus, low-level coffee consumption does not seem to strongly influence the occurrence of GERD, while higher levels of consumption might somewhat increase the risk. However, there is clearly no consensus in the literature about the relationship between coffee consumption and the risk of GERD. Based on a recent meta-analysis and a recent review, GERD symptoms vary strongly between countries, with geographic, ethnic, and cultural characteristics leading to differences in diet and consumption of fried, fatty, sour, and spicy foods that are prone to reflux induction. GERD symptoms increase with age, especially in individuals ≥50 years, and with tobacco and NSAID use, and are more marked in overweight/obese subjects and people with limited physical activity [71,72]. Hence, the factors favoring GERD occurrence are multiple and largely interfere with the potential sensitivity to coffee effects, which do not appear as a main factor in this disorder. However, one clear cause is the stimulatory action of coffee on the esophagus. In one study 66 patients that suffered esophageal pain were as sensitive to any acidic drink, including coffee, as they were to hydrochloric acid [73]. Similarly, LES pressure is significantly reduced by coffee and tea drinking and by increases in GERD [38]. Moreover, the population gathered for each individual study did not drink a homogenous sample of coffee. The type of coffee consumed, its brand and concentration, the way it was processed and prepared, and its composition are unknown and vary among the different studies, countries, and continents. Most often, the studies do not indicate whether the coffee is caffeinated or decaffeinated. 

In this review, we found 15 studies with a lack of association between coffee and GERD, two with a protective effect, and 11 with an aggravating effect. The high variability in the response to coffee in individuals clearly underlies multifactorial causes. Individuals with a negative effect of coffee consumption on GERD might limit or avoid drinking coffee, and on the opposite side subjects with no symptoms may drink more coffee, creating an imbalance in the analysis. Moreover, the existence of esophagitis might influence coffee intake, although most studies did not include endoscopy to detect underlying esophagitis. The questionnaires used in the different studies were not always validated. Furthermore, quite importantly, most studies were cross-sectional and not prospective, which would allow to establish a validated relationship between coffee consumption and GERD. GERD is also affected by body weight, which lowers LES, and GERD symptoms can be resolved by weight loss [74]. In addition, the type of coffee consumed varies; coffee was processed differently in the studies and was often drunk after or with a meal, which can affect the severity of the symptoms. 

In conclusion, this review points to the heterogeneity of the outcomes in the study of the effects of coffee consumption on GERD and to the fact that prospective, well controlled studies are needed to establish a better assessment of the effect of coffee on GERD.

### 4.3. Risk of Peptic Ulcers

Gastric lesions develop upon local dysfunction of homeostasis in the stomach, but there is also a substantial contribution of centrally mediated processes. Indeed, the latter are prominent contributors to some stress-induced gastric lesions. Adenosine A1 receptor agonists induce stomach lesions in rats in the absence of stress and are also able to potentiate the effects of stress [75]. This study did not look at the effects of adenosine receptors antagonists such as caffeine, but these data may allow considering possible beneficial effects of caffeine resulting from its central effect.

Indeed, in most studies, there was no association between coffee drinking and the risk of peptic ulcers. A large meta-analysis of risk factors of peptic ulcers reported that about 90% of peptic ulcer-related symptoms may relate to NSAID use, infection by *H. pylori*, and tobacco smoking [76]. Similar risk factors were reported in a more recent study in the Danish population [77], and coffee does not appear to significantly affect the risk of peptic ulcers. A Japanese cross-sectional study [56] found no effect of coffee consumption on any gastric related disease including gastric and duodenal ulcers in 5451 coffee consumers, compared to 2562 non-consumers from a healthy population with no ulcer history. For the consumption of 1–2 cups/day and ≥3 cups/day compared to consumers of less than 1 cup/day, the odds ratio of gastric ulcer was reduced by 32% (OR = 0.68; 95% CI = 0.29–1.52) and increased by 26% (OR = 1.26; 95% CI = 0.62–2.61), respectively. For duodenal ulcer and at the same levels of intake, the relative risk was reduced by 46% (OR = 0.54; 95% CI = 0.21–1.28) and 21% (OR = 0.79; 95% CI = 0.35–1.80), respectively. A couple of earlier studies did not report any association between caffeine, caffeine-containing beverages, or decaffeinated coffee and the risk of duodenal ulcer [78,79].

In individuals suffering from gastro-esophageal pathologies or ulcers, one study reported that the prevalence of induction of dyspeptic symptoms by coffee was similar in duodenal ulcer patients (29%) and controls (22%) but was much more common in non-ulcer dyspepsia patients (53%) than in controls (22%), *p* = 0.0036. There was no difference in the amount of coffee consumed between patients with duodenal ulcer, non-ulcer dyspepsia, or normal controls [27]. However, there are differences related to ethnicity and/or cultural habits. Thus, duodenal ulcer disease is associated with a high coffee consumption much more frequently among the Japanese population, where this habit is much less prevalent than among the Dutch population, in which there was no association [80].

According to a study in mice, chlorogenic acids could display protective action on gastric mucosa by reducing the surface of the mucosa injured in an experimental ulcer model. Chlorogenic acids do not modify gastric acid secretion but inhibit the migration of the neutrophils involved in the immune response and stimulate the protective action of the enzymes of the anti-oxidative cascade. The authors suggest that the presence of a high level of chlorogenic acids in coffee could explain the lack of effect of coffee on stomach ulcers [81]. Polysaccharides could also play a protective role on gastric mucosa, as emphasized recently in a review on the topic, but no direct study on polysaccharides in coffee and gastric mucosa are available [82]. However, adenosine A2A receptor agonists were reported in rats to block stress-induced gastric inflammation and damage [83]. Another study reported the tonic activity of endogenous adenosine binding to A1 receptors in the formation of stress-induced gastric lesions, while caffeine was protective in the same model [75]. A more in-depth study of the potential action of the effects of caffeine and other coffee components such as different chlorogenic acids and polysaccharides and their interactive/synergistic or antagonist effects on gastric mucosa pathology would need further exploration.

## 5. Bile and Pancreas Secretions Are Stimulated by Coffee

The first part of the small intestine, also called duodenum, is the place where the food bowl or chyme coming out of the stomach is mixed with bile and with pancreatic and intestinal secretions for further digestion steps. Regular coffee and decaffeinated coffee stimulate the secretion of cholecystokinin (CCK), a hormone that stimulates gallbladder function and contractility, and increases the production of bile. This leads to the opening of the Oddi sphincter and allows bile and pancreatic secretion to flow into the small intestine. An increase in the contractility of the gallbladder and a 30% decrease in its volume were found after ingestion of 165 mL of regular or decaffeinated coffee [84].

The stimulation of CCK production is related to caffeinated coffee consumption and to a lesser degree to decaffeinated coffee. CCK is also able to stimulate exocrine pancreas secretion rich in enzymes catalyzing the digestion of lipids, proteins, and carbohydrates [19].

Finally, coffee consumption has been linked to a reduction in the risk of pancreatitis, an inflammation of the pancreas mainly induced by alcohol. The mechanism most likely involved is the protective effect of polyphenols and the diterpenes, kahweol, and cafestol contained in coffee [85], as well as the inhibition of abnormal Ca^2+^ signals due to elevated alcohol consumption [86].

In this aspect of digestion, there are only very few data available, and research still needs to be done as epidemiologic, experimental, and mechanistic studies.

## 6. Coffee Consumption Reduces Gallbladder Stone Formation

Gallbladder stone disease is a common disease affecting 10–15% individuals. Most gallbladder stones are composed of crystallized cholesterol; the rest is solid bilirubin. About 80% of affected individuals are asymptomatic. In Europe, the prevalence ranges from 6 to 22%, with the highest rates in Norway and the lowest in Italy, possibly as a result of differences in diet and obesity rates [87]. The prevalence of gallstone tends to increase in both Europe and the USA [88].

As shown in Table 2, a recent meta-analysis including one case-control and five prospective cohort studies, 227,749 participants, and 11,477 gallbladder stones cases reviewed the association between coffee intake and the risk of gallbladder stones. In prospective cohort studies, coffee intake was significantly associated with a 17% risk reduction (RR = 0.83; 95% CI = 0.76–0.89) of gallbladder stones, but only in women, not in men. In the single case-control study, there was no association between coffee and gallbladder risk in both sexes (OR = 0.99; 95% CI = 0.64–1.53). In the dose-response analysis, the RR of gallstone disease was 5% for the consumption of 1 cup/day of coffee compared to low levels. The risk reduction increases with coffee dosage in a non-linear manner, 5% for 1 cup/day, 11% for 2 cups/day, 19% for 4 cups/day and 25% for 6 cups/day compared to the lowest levels of consumption [89]. Another analytical review that explored the role of several food items on gallstones found a risk reduction of gallstone disease with coffee consumption, although this result was not fully consistent across the studies reviewed [90].

We found 11 individual studies on the relation between coffee consumption and risk of gallstone disease. Among those studies, four studies did not report any association between coffee consumption and the risk of gallstone formation [91,93,94,98]. Two studies found sex-related differences with no effect of coffee consumption on gallstones in men and a tendency [96] or a significant decrease in women [100]. The other five studies found a risk reduction of gallstones with coffee consumption [92,95,97,99,101]. Two studies found dose-related decreases in the risk of gallstones [89,100]. Gallstones occur more frequently in women. Two studies looked at differences between men and women in the association between coffee consumption and gallstone formation [96,100]. They both reported a lack of association in men and risk reduction in women. In the first study [98], the authors reported a 9–13% risk decrease in all women and a 22–24% risk decrease in women with previously diagnosed gallbladder disease for coffee consumption ranging from 1 to ≥2 cups daily. However, the differences were not statistically significant. The second study also found no risk difference in men and that risk decreases were particularly marked in premenopausal women (reduction of 36–83% for 2–≥6 cups coffee daily) and in postmenopausal women that were current users of hormonal replacement therapy (HRT) (reduction of 12–42% for 2–≥6 cups coffee daily). In postmenopausal women who were past users or never had been users of HRT, the risk was unchanged [100]. Finally, one study found coffee-related risk of a decrease in gallstone formation only with caffeinated coffee but not with decaffeinated coffee, hence reflecting the potential protective effect of caffeine [97]. This was confirmed in another study by the same group reporting no difference in the risk reduction effects between caffeinated coffee and caffeine [92].

The mechanism of action underlying the effect of coffee on gallbladder stones is not fully explained. A large part of this effect is attributed to caffeine, as mentioned above. Caffeine, although tendentially promoting increases in cholesterol levels [102], seems to prevent cholesterol crystallization by possibly inhibiting gallbladder absorption [96,103,104]. Caffeine and other methylxanthines may decrease bile cholesterol saturation by stimulating ileal bile acid absorption [105], increasing hepatic bile acid uptake [106]. The reduction of gallbladder stones could also reflect the stimulation of the release of CCK, the enhancement of gallbladder contractility [80], and the improvement of gallbladder mucosal function [103]. All these factors might contribute to the formation of gallbladder stones. However, as can be noticed, most studies cited date back to the 80s’ and 90s’, and more recent research is truly needed to clarify the association and mechanisms of, as well as the components involved in, the potential preventive effect of coffee consumption in gallbladder disease.

## 7. Increase in Colic Motility and Anti-Inflammatory Action

### 7.1. Effects of Coffee on Colon Motility in Physiological Conditions

Coffee does not affect the motility of the small intestine [9] but stimulates colonic motility in 29% of subjects [107]. Distal colonic motility increases as rapidly as 4 min after coffee ingestion [107]. These effects are unlikely to be mediated by caffeine; instead, an indirect action on the colon mediated by neural mechanisms or gastrointestinal hormones has been suspected [108]. Regular and decaffeinated coffee increase the activity of the colon leading to significantly more pressure waves and propagated contractions as compared to water. The activity is especially stimulated in the transverse/descending colon compared to the rectosigmoid portion. Regular coffee stimulates the motility of the colon as much as cereals, 23% more than decaffeinated coffee or 60% more than a glass of water [109]. However, this effect varies with individuals. A study on 99 volunteers reported that 29% described a compelling need to defecate after the ingestion of a cup of coffee, which suggests colonic stimulation. In these individuals, regular and decaffeinated coffee stimulated the motor response at the level of the rectosigmoid portion between 4 and 30 min after ingestion. No stimulation was observed in the other subjects [9]. In another study on eight volunteers, strong coffee (280 mL) increased rectal tone by 45%, while warm water led to a 30% increase. The difference between both treatments was not significant, but both stimulated defecation. None of the drinks induced visceral sensitivity [110]. In another group of healthy volunteers, the consumption of water had no effect on the rectosigmoid motor response, while both regular and decaffeinated coffee induced a motor response in more than half of the volunteers [107]. Coffee consumption is not associated with chronic constipation [111] but rather inversely associated with chronic constipation [112]. The nature of the components of coffee involved in its effects on colon motility has not been studied in detail, but since both regular and decaffeinated coffee are active, caffeine does not appear to be concerned in this effect. The role of other components has not been studied. Some indirect effects have been evoked such as effects of cholecystokinin, gastrin and motilin, whose secretion is stimulated by coffee consumption [2,9,10,19].

### 7.2. Effects of Coffee after Abdominal Surgery

Over the last ten years, the potential preventative role of coffee drinking after abdominal surgery has been tested, since postoperative ileus or bowel paralysis is quite common in the postoperative period after abdominal surgery, such as elective colectomy, colorectal resection, caesarean section, or gynecological surgery. The occurrence of postoperative ileus leads to prolonged patient hospital stay.

Five systematic reviews and meta-analyses on this issue summarized in Table 3 were published recently. Each of them included between four and thirteen randomized control trials concerning both colorectal and gynecological interventions and compared coffee consumption either to water or to no intervention [113,114,115,116,117]. In the five meta-analyses, the time to first flatus was decreased by 3.6 to 10 h. The delay before the first bowel sound was reduced by 3.3 to 12.09 h. The time before first defecation was significantly affected, with reductions ranging from 9.4 to 16.1 h in the coffee group compared to controls. Likewise, the time to tolerance of solid food was highly reduced by coffee by 9.5 to 17.1 h, with the exception of the study of Gkegkes et al. [115], in which only a 1.3 h reduction was reported. Finally, the total length of hospital stay was reduced in all studies by 0.74 to 3.18 days. The lack of complications from coffee was reported by all authors. In gynecologic surgery, there was no significant difference in the risk of postoperative nausea between the participants assigned to the coffee group and those in the control group after cesarean delivery, while a significantly lower risk of postoperative nausea was observed in participants undergoing gynecologic cancer surgery who received coffee compared to the control group. Benefits increased with the increasing complexity of the procedure. None of the studies included reported adverse events or further complications associated with coffee consumption [113]. Coffee was well-tolerated. From all these reviews, it appears that coffee is a low-cost strategy to accelerate postoperative recovery of intestinal function/motility after colorectal and gynecological surgery [114]. Similar data were found in several single studies [118,119,120,121,122,123]. The mechanism of action of these effects is not fully understood, but the significant improvement in gastrointestinal motility due to coffee does not impact postoperative morbidity [115]. 

The most recent review and meta-analysis on postoperative ileus was published at the end of 2021 [117] and included 13 trials concerning colorectal surgeries, cesarean sections, and gynecologic surgeries on a total of 1246 patients. The data from this review are in agreement with the previous ones, and in addition, this review reported a statistically significant reduction of the occurrence of postoperative ileus (RR = 0.42; 95% CI, 0.26–0.69). Likewise, recent studies from single centers confirmed the previous observations from the meta-analyses and older single studies. Administration of coffee significantly decreased the duration to first defecation both after gynecological surgery and cesarean section [124,125] and after elective colorectal resection [126]. In addition, after cesarean section, coffee shortened the mean time to the first flatus and the duration before tolerance of solid food compared to control values [125].

The mechanism of action of these effects is not fully understood, but the significant improvement in gastrointestinal motility due to coffee does not impact postoperative morbidity [115]. One study reported an effect similar for decaffeinated or regular coffee [122], indicating that the effect is rather linked other compounds such as chlorogenic acids and melanoidins [2].

More recent studies from single centers confirmed the previous observations. Administration of coffee significantly decreased the duration to first defecation both after gynecological surgery and cesarean section [124,125] and after elective colorectal resection [126]. In addition, after cesarean section, coffee shortened the mean time to the first flatus passage and the duration to tolerance of solid food compared to control values [125].

### 7.3. Effects of Coffee on Inflammatory Bowel Disease

Inflammatory bowel disease (IBD) comprises various inflammatory diseases of the intestine such as Crohn’s disease and ulcerative colitis. Over recent years, the consequences of nutritional intake and IBD have been explored. According to a recent study comprising 442 patients with Crohn’s disease or ulcerative colitis, it appeared that 73% regularly consume coffee. Almost the same percentage, 96% of patients considering a positive action of coffee and 91% reporting no impact of coffee intake on IBD, consumed coffee daily. Among the patients refraining from regular coffee intake, 62% were convinced that coffee adversely influences intestinal symptoms, significantly more in Crohn’s disease than in ulcerative colitis patients (76% vs. 44%, *p* = 0.002). In total, 38% of all subjects considered coffee to affect their symptoms, significantly more in Crohn’s disease (54%) than in ulcerative colitis patients (22%, *p* < 0.001). However, this negative perception did not translate into abstinence from coffee consumption [127].

The results on this pathology appear discordant. Several studies reported that patients with Crohn’s disease and ulcerative colitis tend to consume less coffee than controls [128], and that consumption of coffee was causing or aggravating symptoms [129,130,131,132] or was a strong predictor of the occurrence of IBD [132]. On the other hand, IBD was not associated with coffee in one Turkish study [133], while coffee was even reported to be protective (adjusted OR = 0.51; 95% CI = 0.51–0.98) in a study on ulcerative colitis [134]. A Mendelian randomization study found no evidence that genetically predicted coffee consumption could be causally linked to the occurrence of IBD [135]. Similarly, a meta-analysis of 16 epidemiological observational studies on the association between beverage intake and Crohn’s disease, including five studies on coffee, found no association between coffee consumption and the risk of developing Crohn’s disease (RR = 0.82; 95% IC = 0.46–1.46). Conversely, high intake of carbonated soft drinks increased the risk while tea was protective. Coffee might play various roles in the etiology and manifestation of the disease. In case of mucosa inflammation, coffee could be protective, but its role in the intestinal tract before the expression of the disease might be variably affected by a wide array of factors [136].

## 8. Coffee Influences the Composition of the Intestinal Microbiota

Two major phyla, *Firmicutes* and *Bacteroides*, represent about 90% of the bacteria identified in the gastrointestinal tract [137]. The gut of healthy subjects harbors mainly three types of enterobacteria—*Bacteroides, Prevotella*, and *Ruminococcus* [138]—but a large inter-individual variability has been reported in microbiota composition [137].

Dietary fibers that are mostly nondigestible polysaccharides contained in coffee have marked effects on gut microbiota. They are rapidly metabolized into short-chain fatty acids and cause an increase of up to 60% of the levels of the *Bacteroides/Prevotella* bacteria group after fermentation by human fecal samples in the presence of medium roasted Arabica [139]. The same group also reported that in the presence of Arabica, the increase in the same bacterial group ranged from 2–40% after fermentation depending on the molecular weight of the fraction and the degree of roasting (light, medium, dark) of the coffee [140].

Initially, a Swiss group looked at the effect of coffee on the gut microbiota in 16 adult healthy subjects after the consumption of 3 cups of coffee per day for 3 weeks. Although fecal profiles of the dominant microbiota were not significantly affected after coffee consumption, the population of *Bifidobacterium* spp. increased after the 3-week test period. Moreover, in some subjects, there was a specific increase in the metabolic activity of *Bifidobacterium* spp. These data show that the consumption of the coffee preparation resulting from water coextraction of green and roasted coffee beans produces an increase in the metabolic activity and/or numbers of the *Bifidobacterium* spp. population, a bacterial group of recognized beneficial effects, without major impact on the dominant microbiota [141]. Significant increases in the growth of *Bifidobacterium* spp. relative to controls were also reported after 10 h incubation of coffee samples enriched in chlorogenic acid with human fecal distal colon samples. Chlorogenic acid alone had a similar effect and also induced a significant increase in the growth of the *Clostridium coccoides-Eubacterium rectale* group. Fermentation with a concentration of chlorogenic acids equivalent to a coffee rich in chlorogenic acids produced a similar stimulation of the growth of *Bifidobacteria* spp. [142], reflecting that the effects of coffee on gut microbiota seem to be mainly due to its richness in polyphenols [142,143]. A recent intervention study reported that the ingestion of 200 mg caffeine along with 200 mg chlorogenic acids increased the population of *Bifidobacteria*, while neither caffeine nor chlorogenic acids provided separately achieved this effect [144]. Likewise, in mice, green tea extract supplementation after stress accelerated the recovery of the populations of *Bifidobacteria* and *Lactobacillus* spp. [145]. Thus, the selective metabolism and subsequent amplification of some bacterial populations upon coffee consumption could be beneficial, but the consequences of these changes are not fully understood yet.

An extract of medium-roasted Arabica coffee promoted the growth of *Bifidobarium animalis* along with *Lactobacillus rhamnosus* and *Lactobacillus acidophilus* and dark-roasted Arabica coffee extract stimulated the growth of *Bifidobacterium animalis* ssp. *lactis* BB12. This effect was also produced by higher polysaccharide and chlorogenic acid contents, but not by caffeine that did not stimulate the growth of *Bifidobacterium* spp. and only mildly stimulated the growth of *Lactobacillus* spp. On the other hand, the same study reported that regular Arabica and Robusta coffees inhibited the development of the pathogenic bacteria *Escherichia coli*, while decaffeinated coffee stimulated its growth [146]. Another study found higher levels of *Bacteroides/Prevotella/Porphyromonas* in heavy coffee consumers (45–500 mL/day) who also showed lower peroxidation activity. This effect is attributed to both polyphenols, mainly chlorogenic acids and caffeine that are present in coffee [147]. Another species of *Bacteroides*, i.e., *Bacteroides thetaiotaomicron* was predominant along with *Fascolartobacterium faecium* and *Eubacerium rectale* in subjects who consumed coffee [148].

When considering individuals with several pathologies, coffee polyphenols were reported to lead to increases in the population of *Bacteroides plebeius* and *Bacteroides coprocola* in hypertensive subjects, while they decreased the population of *Faecalibacterium prausnitzii* and *Christensenellaceae* R-7 in normotensive individuals [149]. Another study reported in 23 patients with allergies that coffee rich in polyphenols stimulated the growth of species of *Clostridium*, *Lactococcus* and *Lactobacillus* [150]. In vivo, coffee caused an increase in the anti-inflammatory *Bifidobacteria* population, a decrease in *Clostridium* spp., linked to colon inflammation and to *Escherichia coli*, which is usually innocuous but potentially pathogenic and which may invade the gut mucosa in inflammatory states as well as the gut of patients with Parkinson’s disease [141,151,152,153].

In constipated subjects, decreased *Prevotella* abundance and a more consistent decrease in abundance of *Bifidobacteria* was found [154,155,156]. In patients with IBD compared to healthy controls, a caffeine consumption of at least 400 mg/day increased bacterial diversity as well as the abundance of some species, such as *Parabacteroides, Oscillibacter, Lachnospiraceae*-unclassified, and *Ruminococcaceae*-unclassified [157]. In male Tsumura Suzuki obese diabetes mice, a spontaneous mouse model of metabolic syndrome, daily intake of coffee, caffeine or chlorogenic acid for 16 weeks prevented nonalcoholic hepatic lobular inflammation and changed the abundance of various species of the gut flora, such as *Coprococcus/Blautia*/*Prevotella* involved in the regulation of inflammation [158]. In a model of metabolic syndrome in rats, supplementing the diet with spent coffee grounds for 8 weeks improved glucose tolerance and structure/function of liver and heart. Spent coffee grounds increased the richness/diversity of gut microbiota and decreased the ratio of *Firmicutes*-to-*Bacteroidetes* [159]. In a pathological model of obesity, the administration of caffeinated coffee for 10 weeks to high-fat-fed rats was associated with decreased body weight, adiposity, liver triglycerides, and energy intake. Coffee consumption attenuated the increase in *Clostridium Cluster XI* normally associated with high-fat feeding and resulted in augmented levels of *Enterobacteria* [160].

Microbial communities within the oropharynx are grouped into three distinct habitats, which themselves showed no direct influence on the composition of the gut microbiota [161]. One single study reported that coffee exposure increases only the population of *Granulicatella* and Synergistetes at the level of the oral microbiota [162].

It is not easy to conclude on the effects of coffee consumption on the composition and function of gut microbiota and its general consequences on health because of the diversity of the data presently available. Excellent information can be found in a recent Spanish paper detailing the mechanisms of action of coffee on the gastrointestinal tract. The authors reviewed the effects of coffee on the different layers of the gut wall, and the relations between gut and brain [2]. Many more studies will be needed to better clarify the impact of coffee intake on the composition of gut microbiota and its potential consequences on health. Indeed, the composition of the microbiota seems to be linked to several health disorders such as inflammatory bowel disease, non-alcoholic liver steatosis, cardiovascular diseases, diabetes, obesity, cancer, and Parkinson’s disease. Coffee was reported to have beneficial actions on these pathologies. 

## 9. Coffee and Cancer of the Gastrointestinal Tract

The main purpose of this review was to gather the information available on the effects of coffee on the physiology of the gastrointestinal tract. We will just evoke briefly the consequences of coffee consumption on cancers at different levels of the digestive system.

In 2016, a panel of experts from the International Agency for Research on Cancer (IARC) re-assessed the effect of coffee consumption on cancer incidence. After reviewing more than 1000 studies in humans and animals, the Working Group concluded that there was no evidence supporting the carcinogenicity of coffee drinking overall, and classified coffee in Group 3, i.e., “not classifiable as to its carcinogenicity to humans” [163,164]. With the exception of the liver, which will be treated separately, for most digestive cancers, the IARC Expert group found the evidence of a relation between coffee consumption and digestive cancers to be inconsistent. Only for pancreatic cancer did they state that there was a lack of association between this cancer and coffee consumption, based on a large number of cohort and case-control studies [163,164]. The association between coffee exposure and digestive cancers has been further studied in many recent studies and meta-analyses, and this review will comment only briefly on studies that were performed after the IARC conclusions were published.

### 9.1. Cancer of the Oral Cavity and Esophagus

Coffee intake reduces the risk of oral cavity cancer at all doses, in both case-control and cohort studies, as reported by a meta-analysis of 15 studies [165]. No association between coffee consumption and esophageal cancer was reported in a recent meta-analysis of 11 studies. An inverse association was found in East Asian subjects with an OR of 0.64 (95% CI = 0.44–0.83), but not in European and American subjects (OR = 1.05; 95% CI = 0.81–1.29) [166]. There was some doubt about esophageal cancer, since this cancer has been often observed in subjects drinking very hot beverages, mainly tea and mate [163]. The preferred temperature for the consumption of coffee has been found to be 60 ± 8.3 °C, while the optimum temperature safe for the esophageal mucosa is considered as being 57.8 °C [167]. Hence, the usual temperature of coffee consumption is lower than the temperature leading to burns and lesions of the esophageal mucosa with potential evolution to cancer [168].

### 9.2. Cancer of the Stomach and Pancreas

Stomach cancers are classified as cardia and non-cardia type depending on their anatomical location. Cardia subtypes behave as esophageal cancers, while non-cardia types most often relate to the presence of *Helicobacter pylori* in addition to risk factors common to both types [1]. A recent meta-analysis of 22 studies, including 9 cohort and 13 case-control studies involving 7631 cases, reported that coffee intake, at any level, significantly decreases the risk of developing stomach cancer by 7% compared to no consumption. The risk reduction reached 12% in high consumers (3–4 cups/day) and 5–8% in consumers of less than one or 1–2 cups/day [169]. A pooled analysis including 18 studies, 8198 cases, and 21,419 controls reported no risk change in stomach cancer with an estimated OR of 1.03 (95% IC = 0.94–1.13) for coffee drinkers compared to rare or non-drinkers. When considering the amount of coffee consumed, the pooled ORs reached 0.91, 0.95, and 0.95 for 1–2, 3–4 and ≥5 cups daily. A high consumption of ≥7 cups of coffee daily increased the risk of stomach cancer by 20%. A positive association was found for the intake of ≥5 cups of coffee daily and gastric cardia cancer [170]. An older meta-analysis of prospective cohort studies including 3484 cases from 1,324,559 participants reported a 50% increased risk in stomach cardia cancer, especially in American populations. This increased risk disappeared after adjusting for smoking and body mass index [171].

A recent systematic review and meta-analysis on the association between coffee and the risk of pancreatic cancer included 13 high-quality cohort studies and involved 3831 cases among 959,992 participants. The pooled relative risk of developing pancreatic cancer reached 1.08 (95% CI = 0.94–1.25) in the highest compared to the lowest consumption category. The increment of each cup of coffee increased the risk by 6%. The authors concluded that there was an association between the risk of pancreatic cancer and coffee consumption [172]. On the opposite side, a meta-analysis of one cohort study in 309,797 never-smoker women with a 13.7-year follow-up did not find any statistically significant association between coffee consumption and pancreatic cancer risk with a summary RR = 1.00 (95% CI = 0.86–1.17) for ≥2 versus zero cups of coffee/day. The risk ratios for pancreatic cancer in this group of women regularly decreased, reaching 1.02 (95% CI = 0.83–1.26), 0.96 (0.76–1.22), and 0.87 (0.64–1.18) for the daily consumption of 1–2, 3–4, and ≥5 cups of coffee compared to non-consumption, respectively [173]. Other prospective studies and meta-analyses also reported a decreased risk of pancreatic cancer associated with high coffee consumption [174]. Of note also, heavy coffee consumption was associated with decreased risk of pancreatitis [175]. Additional studies are still needed to clarify the relationship between coffee consumption and pancreatic cancer.

### 9.3. Cancer of the Gallbladder

Concerning gallbladder cancer, there are only very few studies available. Recently, in a cohort of 72,680 Swedish adults (45–83 years), 74 gallbladder cancer cases were identified. Compared with the daily consumption of one cup of coffee at the most, the risk of developing gallbladder cancer was decreased by 24%, 50%, or 59% for the consumption of 2 cups, 3 cups, or ≥4 cups per day. These data reflect a strong inverse association between coffee consumption and gallbladder cancer risk [176].

### 9.4. Colorectal Cancer

There are numerous studies on the relationship between coffee consumption and colorectal cancer. As in previous reports, the association is variable. One review and meta-analysis of prospective studies found a protective effect of coffee against colon cancer in men and women combined and in men alone. The effect was mostly significant in European men and Asian women. There was no association between coffee and rectal cancer [177]. In another meta-analysis limited to nine prospective cohort studies from Asia, the authors found an inverse association between coffee and colon cancer risk with a summary RR of 0.90 (95% CI = 0.79–1.03) in men and 0.64 (95% CI = 0.36–1.15) in women [178]. Finally, the Cancer Prevention Study-II Nutrition Cohort included 47,010 men and 60,051 women leading to 1829 colorectal cases over 12 years of follow-up. There was no association between caffeinated coffee consumption and the risk of colon cancer (hazard ratios (HRs) ranging from 0.90–0.95) or possible risk reduction in proximal colon cancer (HRs ranging from 0.72–0.77). On the other hand, the risk increased with the consumption of caffeinated coffee in distal colon cancer (HRs ranging from 1.20–1.37) and in rectal cancer (HRs ranging from 1.00–1.47). On the contrary, the same study reported that consumption of decaffeinated coffee reduced the risk of cancer at all colonic sites, with HRs ranging from 0.78–0.99 and reaching 0.63–1.07 for cancer of the rectum. These differences with locations and reduced effects in distal colon and rectum are not yet understood [179] and might explain the variability in study outcomes. In addition, coffee consumption was reported to reduce the risk of disease progression and death in patients suffering advanced or metastatic colorectal cancer [180].

### 9.5. Cancer of the Liver

Chronic liver disease is the fifth most frequent cause of death. Likewise, liver cancer is the fifth most common cancer in men and the ninth most frequent in women worldwide. The recent analysis by the experts of the IARC on the relation between coffee intake and the incidence of liver cancer reported inverse associations in cohort and case-control studies, and in meta-analyses concerning populations from North America, Europe, and Asia [163,164]. A meta-analysis of twelve recent prospective cohort studies including 3414 cases found a summary relative risk of 0.66 (95% CI = 0.55–0.78) for regular, 0.78 (95% CI = 0.66–0.91) for low, and 0.50 (95% CI, 0.43–0.58) for high coffee consumption. The meta-analysis reported also that each additional cup of coffee reduces the risk of liver cancer by 15% [181]. Several other meta-analyses reached similar conclusions. A meta-analysis of eighteen cohort and eight case-control studies reported an inverse association between coffee consumption and liver cancer with a risk ratio of 0.71 (95% CI = 0.65–0.77) for cohort studies and of 0.53 (95% CI = 0.41–0.69) for case-control studies, but the former are considered of higher quality than the latter. This association was not altered by the stage of the disease, the level of alcohol consumption, elevated body mass index, type 2 diabetes, or hepatitis B or C. Both caffeinated and decaffeinated coffee are active [182]. Another meta-analysis of 20 case-control and prospective cohort studies found a mean decreased risk of liver cancer by coffee consumption of 0.69 (95% CI = 0.56–0.85). In this meta-analysis, the authors reported that in several of the studies included, a significant protection against liver cancer occurrence became significant at a coffee consumption of 2 cups/day [183]. In all these studies, the degree of protection increased with larger amounts of coffee consumed. A meta-analysis of six Japanese studies reported a large decrease in the relative risk of developing liver cancer in coffee consumers, reaching a value of 0.50 (95% CI = 0.38–0.66) [184]. The additional consumption of two cups of coffee daily was reported to decrease the relative risk of developing liver cancer by 27% (RR = 0.73; 95% CI = 0.63–0.85) in one meta-analysis [2] and by 43% in another study (RR = 0.57; 95% CI = 0.49–0.67) [184]. In a further analysis, the latter authors found a 31% decreased risk for subjects without a history of liver disease (RR = 0.69; 95% CI = 0.43–0.58) and 44% for persons with a history of liver disease (RR = 0.56; 95% CI = 0.35–0.91) [185]. Finally, coffee consumption was also reported to be associated with a reduced risk of liver cancer recurrence and to prolong survival in patients that underwent orthotopic liver transplantation [186].

A recent study from the UK Biobank also reported that regular, instant, or decaffeinated coffee reduces the risk of chronic liver disease. The latter disease includes viral hepatitis B and C, alcohol-induced liver disease or cirrhosis, and non-alcoholic fatty liver disease (NAFLD). In this large study involving 348,818 coffee consumers and 109,767 non-coffee consumers, the authors observed 3600 cases of chronic liver disease, 5439 cases of liver steatosis, 184 cases of liver cancer, and 301 deaths over a median 10.7-year follow-up. Compared to non-consumers, the risk of developing chronic liver disease, steatosis, and cancer was reduced by 20%, while the risk of death from a liver pathology was reduced by 49% in coffee consumers. This reduction was found with all types of coffee, regular caffeinated ground coffee, decaffeinated coffee, and instant coffee [187]. Likewise, another relatively recent cohort study including 1019 patients reported that high coffee consumers had a lower risk of advanced liver fibrosis than the reference group. The adjusted OR reached 0.14 (95% CI = 0.03–0.64) in high-risk alcohol drinkers and 0.11 (95% CI = 0.05–0.25) in low-risk alcohol drinkers. The authors concluded that elevated coffee consumption is associated with reduced risk of liver fibrosis, even in HIV-HCV co-infected patients with elevated risk alcohol consumption [188]. All studies devoted to the effects of coffee consumption in patients with NAFLD concluded that there is a protective effect of coffee against liver fibrosis. A recent meta-analysis of 11 studies reported that coffee intake reduces the risk of developing NAFLD in coffee consumers compared to non-consumers by 32% (RR = 0.68; 95% CI = 0.68–0.79), but also reduces the aggravation of the disease in already diagnosed patients by 23% (RR = 0.77; 95% CI = 0.60–0.98) [189]. The results of another very recent meta-analysis including five studies from 2011 to 2016 suggest that a more elevated coffee intake is inversely associated with the severity of hepatic fibrosis in patients with NAFLD. The optimal quantity as well as the type of coffee and the preparation to reach these effects remain unknown [190].

Coffee consumption acts also on the circulating level of the liver enzymes aspartate aminotransferase (AST) and alanine aminotransferase (ALT). As reported recently in a meta-analysis of 19 observational studies, coffee intake was inversely related to elevated levels of ALT (RR = 0.69; 95% CI = 0.60–0.79) and AST (RR = 0.62; 95% CI = 0.48–0.81) [191]. A US study was performed on 5944 adults of the Third US National and Nutrition Examination Survey, 1988–1994, characterized by overweight, impaired glucose metabolism, iron overload, viral hepatitis, and excessive alcohol consumption. Their ALT activity was high (>43 U/L), indicating liver injury. In subjects drinking over two cups of coffee daily compared to abstainers, the odds ratio reached 0.56 (95% CI = 0.31–1.00), reflecting a protective effect of coffee against liver injury [192].

Finally, coffee has also been reported to protect against viral hepatitis. A US study included 766 participants infected by hepatitis C followed-up for 3.8 years. At baseline, patients consuming coffee had less severe liver steatosis and a lower serum AST/ALT ratio. Outcomes occurred in 230 patients and their frequency progressively declined with increasing coffee consumption from 11.1/100 person-years for no coffee down to 6.3 for a consumption ≥ 3 cups/day. The RRs were 1.11 (95% CI = 0.76–1.61) for <1 cup/day, 0.70 (95% CI = 0.48–1.02) for 1 to <3 cups/day and 0.47 (95% CI = 0.27–0.85) for ≥3 cups/day compared to no consumption. These data show that coffee consumption is associated with reduced rates of liver disease progression [193]. Similar protective data were found in a meta-analysis concerning fibrosis in patients infected with the hepatitis C virus. In the five studies included, the pooled OR of advanced liver cirrhosis in HCV patients who consumed caffeine regularly compared to an absence of consumption reached 0.48 (95% CI = 0.30–0.76), and the pooled OR for moderate/severe histological inflammation reached 0.61 (95% CI = 0.35–1.04) [194]. The consumption of two cups of coffee was also reported to decrease liver stiffness in 155 patients with NAFLD, 378 with HCV, and 485 with HBV, independently from disease stage. This may reflect reduced fibrosis and inflammation [195].

The mechanisms underlying these powerful effects of coffee in all types of liver disease will not be detailed here. Two excellent reviews among others detail the participation of caffeine, trigonelline, chlorogenic acid, and melanoidins among other components in the hepatoprotective effect of coffee [2,196].

## 10. Conclusions

Since coffee is widely consumed worldwide, it is of critical importance to know its effects on the first organs of the body with which it comes in contact during consumption, i.e., the gastro-intestinal tract. Surprisingly, research devoted to this aspect remains scarce. The data reviewed here show that coffee intake stimulates gastric, biliary, and pancreatic secretions, seeming to favor the first steps of the digestive process. Most data are not in favor of a direct effect of coffee on gastro-esophageal reflux, which is rather a combined or additive effect to other risk factors such as obesity and a poor diet. Coffee stimulates the motor activity of the colon, and its use is now recurrently suggested as a non-harmful adjuvant to restore colonic contraction and function after abdominal surgery. Coffee consumption induces changes in the composition of the gut microbiota, mainly at the level of the population of *Bifidobacteria, Bacteroides*, and *Prevotella*. Coffee consumption has not been reported to generate any deleterious effects on the various organs of the digestive tract, and its prominent protective effects against hepatic carcinoma and all other liver diseases have been largely reported. Nevertheless, additional data are greatly needed, since, at various steps of the digestive process, mostly only quite old data are available and they are heterogeneous given the variability of the type of coffee use, the way it was processed, its concentration, and its mode of preparation. Further prospective studies would be needed with modern technology applied on larger population groups.

## Figures and Tables

**Table 1 nutrients-14-00399-t001:** Coffee and gastro-esophageal reflux (GERD).

Authors	Study Design	Country	Size of the Population	Age (Years)	Number of Cases and Controls	Outcome
Reviews and meta-analyses						
Kim et al., 2014 [47]	Meta-analysis of 15 case-control studies between 1999 and 2012	Europe America Asia	113 to 43,363 participants per study included			No association between coffee consumption and GERD: <4 cups/day: OR = 0.91 (95% CI = 0.81–1.01) >5 cups/day: OR = 1.14 (0.69–1.88)
Chen et al., 2021 [48]	Meta-analysis of 21 retrospective, prospective, Asian, and high-quality studies	Europe America Africa Asia	24,943 participants	16,297 adults, 7299 adolescents		*No significant association between coffee and GERD in all 21 studies* *pooled*: (RR = 1.07; 0.96–1.19; *p* < 0.001) *No significant association in* - *Retrospective studies:* RR = 1.05 (0.91–1.22; *p* = 0.515) - *Asian studies:* RR = 1.08 (0.96–1.21; *p* = 0.014) - *Questionnaire-based studies:* RR = 1.1 (0.84–1.21; *p* = 0.004) - *High-quality studies:* RR = 1.01 (0.91–1.12; *p* < 0.001) - *Prospective studies:* RR = 1.13 (0.97–1.32; *p* = 0.015) *Significant association between coffee and GERD in cross-sectional studies:* RR = 1.54 (0.40–5.93; *p* < 0.001)
Single studies: GERD						
No association between coffee and GERD						
Chang et al., 1997 [42]	Case-control study	China	2044 endoscopy patients	16–82 years	102 patients with GERD, 1932 without GERD, 1266 males, 778 females	*No association between coffee consumption and the risk of GERD*
Boekema et al., 1999 [49]	Randomized, controlled crossover study	The Nether-lands	15 subjects	20–61 years	7 cases, 33–50 years 8 controls, 20–61 years	*No effect of coffee on postprandial acid reflux time or number of reflux episodes*, both in GERD patients and in healthy subjects. In the fasting period, coffee increased the percentage acid reflux time only in GERD patients [median 2.6, (0–19.3) vs. median 0 (0–8.3), *p* = 0.028], compared to healthy subjects
Nilsson et al., 2004 [50]	Case-control study	Norway	47,556 participants	19–101 years	3155 subjects with GERD or heartburn (1555 males and 1590 females) 40,120 controls (18,814 males and 21,396 females)	*No association between coffee consumption and the risk of GERD* compared to a daily consumption of less than one cup 1–3 cups/day: OR = 1.0 (0.8–1.1) 3 cups/day: OR = 1.1 (0.9–1.5)
Dore et al., 2007 [51]	Single center case-control study	Italy	500 subjects 169 males 331 females	15–61 years	300 cases 200 controls 390 coffee consumers	*No association between coffee consumption and GERD diagnosed by endoscopy* Age- and sex-adjusted OR for coffee vs. no consumption: OR = 1.0 (0.6–1.2)
El Serag et al., 2007 [52]	Retrospective nested case-control study	USA	113 subjects 48 males 65 females	17–18 years	All were patients diagnosed with GERD in childhood (10–12 years)	*No association between GERD and coffee in adults with history of childhood GERD*
Zheng et al., 2007 [53]	Swedish Twin Registry Survey with questionnaires and telephone interviews	Sweden	23,634 subjects 10,950 males and 12,684 females	57 (42–99) years for males 58 (42–104) years for females	Males: 1753 with GERD, 9197 controls Females: 2330 with GERD, 10,354 controls	*No association between GERD and coffee in female subjects* Adjusted ORs compared to no consumption: 1–3 cups/day: OR = 0.92 (0.76–1.12) 4–6 cups/day: OR = 1.01 (0.82–1.25) ≥7 cup/day: OR = 1.10 (0.85–1.43)
Friedenberg et al., 2010 [54]	Cross-sectional survey	USA	503 subjects 374 controls, 147 males and 227 females 129 with GERD, 129 with GERD	42.3 ± 17.2 years for controls 44.9 ± 15.9 years for controls	374 controls, 129 with GERD	*No association between coffee consumption and GERD* Association between body mass index (high BMI in this population (29.6 ± 9.1) and GERD
Bhatia et al., 2011 [44]	Multicenter case-control study	India	3224 participants	GERD patients: 38.4 (28–48;5) years Controls, 40 (30–52) years	245 cases, 112 males and 133 females 2335 controls, 1534 males and 1444 females	*No association between coffee consumption and GERD* Multivariate OR = 0.66 (0.29–1.50; *p* = 0.309) compared to no coffee
Pandeya et al., 2012 [55]	Cross-sectional survey	Australia	1580 subjects 1040 males 540 females	30–70	727 control subjects; 678 subjects with weekly GERD; 175 subjects with weekly GERD	*No association between coffee consumption and GERD.* Occasional GERD symptoms compared to no coffee consumption ≤3 cups/month: PR = 0.93 (0.73–1.18) 1–6 cups/week: PR = 1.04 (0.85–1.29) ≥1 cup/day: PR = 1.02 (0.82–1.26) Frequent GERD symptoms compared to no coffee consumption ≤3 cups/month: PR = 1.23 (0.69–2.19) 1–6 cups/week: PR = 1.08 (0.63–1.85) ≥1 cup/day: PR = 1.20 (0.70–2.05)
Shimamoto et al., 2013 [56]	Cross-sectional study	Japan	8013 subjects, 5451 coffee drinkers, 2562 non-drinkers	49.8 ± 8.2 years in drinkers and 51.5 ± 9.7 years in non-drinkers	4670 females (3194 drinkers and 1476 non-drinkers) 3343 males (2257 drinkers and 1086 non-drinkers)	*No association between coffee consumption and risk of reflux esophagitis (compared to less than one cup of coffee/day)* 1–2 cups/day, OR = 0.88 (0.74–1.04) ≥3 cups/day: OR = 0.84 (0.70–1.01) *No association between coffee consumption and risk of non-erosive reflux disease (compared to less than one cup of coffee/day)* 1–2 cups/day, OR = 0.93 (0.79–1.08) ≥3 cups/day: OR = 0.93 (0.79–1.10)
Ercelep et al., 2014 [57]	Retrospective nested case-control study	Turkey	2037 subjects 791 males 1246 females	35.9 ± 9.7 (without GERD) 36.8 ± 9.6 (with GERD)	1595 without GERD (636 males, 959 females) 442 with GERD (155 males, 287 females)	*No association of coffee consumption with GERD symptoms*, OR = 1.06 (0.66–1.70) for those drinking more than 3 cups versus non-drinking or drinking less
Kubo et al., 2014 [58]	Case-control study	USA	490 subjects 334 males, 136 females	20–79 years	181 controls (123 males, 68 females) 380 patients with GERD (211 males, 69 females)	*No association between coffee consumption and GERD (compared to no coffee consumption)* ≥2 cups/day: OR = 0.89 (0.52–1.51)
Filiberti et al., 2017 [59]	Retrospective case-control study: patients with esophagitis (E)	Italy	1420 subjects 766 males 654 females	53.7 ± 14.1 years for controls 52.6 ± 14.7 years for E patients	619 controls (252 males, 367 females) 462 E patients (285 males, 177 females)	*No association between coffee and esophagitis in current drinkers*, increased risk of E in light drinkers (<1 cup/day): OR = 1.85 (1.00–3.43) vs. controls
Wei et al., 2019 [60]	Prospective study	Taiwan	1837 subjects, 1197 coffee drinkers and 185 heavy consumers	Whole sample: 51.7 ± 10.2 years Males: 51.7 ± 10.4 years Females: 51.4 ± 10.0 years	970 males and 867 females	*No association between coffee consumption and GERD (compared to no coffee consumption)* Coffee consumption, OR = 1.11 (0.86–1.43) Heavy coffee consumption (>2 cups/day): OR = 0.99 (0.69–1.43) *No association between coffee consumption and erosive esophagitis on endoscopy (compared to no coffee consumption)* *Males* Coffee consumption, OR = 0.86 (0.61–1.22) Heavy coffee consumption (>2 cups/day): OR = 0.86 (0.52–1.43) *Females* Coffee consumption, OR = 0.98 (0.63–1.52) Heavy coffee consumption (>2 cups/day): OR = 1.16 (0.60–2.26)
Yuan et al., 2019 [61]	Multicenter case-control study	China	1518 subjects, 832 GERD patients and 686 controls	GERD: 48.5 ± 13.2 years Non-GERD: 47.5 ± 14.86 years	GERD: 455 males, 377 females Non-GERD: 302 males, 384 females	*No association between a preference for coffee drinking and GERD* OR = 1.27 (0.78–2.05)
Reduced risk of GERD in coffee consumers						
Diaz-Rubio et al., 2004 [62]	Random population sample based on telephone interviews	Spain	2500 subjects, 1185 males and 1315 females	40–79 years	245 subjects with frequent GERD or dyspepsia, 546 subjects with occasional symptoms	*Reduced risk of GERD symptoms and frequency in patients with frequent vs. occasional symptoms* Symptoms vs. no-symptoms: OR = 0.85 (0.67–1.06) Frequent vs. occasional symptoms: OR = 0.66 (0.66–0.97)
Zheng et al., 2007 [53]	Swedish Twin Registry Survey with questionnaires and telephone interviews	Sweden	23,634 subjects 10,950 males and 12,684 females	57 (42–99) years for males 58 (42–104) years for females	Males: 1753 with GERD, 9197 controls Females: 2330 with GERD, 10,354 controls	*Inverse dose-dependent reduced risk of GERD associated to coffee consumption only in in male subjects* Adjusted ORs compared to no consumption: 1–3 cups/day: OR = 0.91 (0.73–1.12) 4–6 cups/day: OR = 0.86 (0.69–1.08) ≥7 cup/day: OR = 0.75 (0.57–0.98)
Increased risk of GERD in coffee consumers						
Wendl et al., 1994 [41]	Interventional double-blinded randomized study	Germany	16 healthy volunteers	25.9 (20–41) years	7 men 9 women	*Association between regular and GERD compared with tap water*. No effects of decaffeinated coffee and tap water. GERD: 3.2% (1.3–14.4%) with regular coffee and 0.9% (0.1–3.6%) with decaffeinated coffee (*p* < 0.05)
Pehl et al., 1997 [63]	Interventional double-blinded randomized study	Germany	17 reflux patients 11 males, 6 females	47–78 years	9 with endoscopic esophagitis 8 controls	*Association between by regular coffee and GERD in both patients with and without reflux oesophagitis* Reduction of this effect by 83% with decaffeinated coffee Median values of fraction time esophageal pH: Caffeinated coffee = 17.9 (0.7–56.6) Decaffeinated coffee = 3.1 (0–49.9), *p* < 0.001
Diaz-Rubio et al., 2004 [62]	Random population sample based on telephone interviews	Spain	2500 subjects, 1185 males and 1315 females	40–79 years	245 subjects with frequent GERD or dyspepsia, 546 subjects with occasional symptoms	*Association between regular coffee, GERD symptoms frequency, and severity in patients with severe or long-lasting GERD* Severe vs. non-severe symptoms: OR = 1.15 (0.58–2.30) ≥10 years vs. less: OR = 1.18 (0.79–1.76)
Wang et al., 2004 [64]	Epidemiologic, based on questionnaires	China	2789 residents	18–70 years	85 responders, 17 with GERD	*Mild association between coffee and GERD* OR = 1.23 (0.76–2.00)
Martin-de-Argila & Martinez-Jiménez 2013 [65]	Multicenter, cross-sectional, retrospective and non-interventional study	Spain	2246 patients with GERD	18–70 years	1002 males 1244 females	*Coffee intake (>1 vs. <1 cup/day) significantly related to chest pain* OR = 1.33 (1.01–1.75)
Park et al., 2014 [66]	Prospective case-control study: patients monitored for upper GI cancer	Korea	2226 subjects	46.3 (19–87) years	742 subjects with GERD (460 males, 282 females) and 1484 healthy controls (920 males, 564 females)	*Association between regular coffee and GERD in patients with and without reflux oesophagitis (RE) for coffee vs. none* RE risk in the whole sample: OR = 1.35 (1.13–1.43) RE symptoms: OR = 1.45 (1.07–1.96) Risk in young group (<40): OR = 1.13 (1.08–1.31) Mean age group (≥40–<65): OR = 1.30 (1.12–1.52) Elderly group (≤65): OR = 1.40 (1.09–2.10)
Alsulobi et al., 2017 [67]	Cross-sectional study	Saudi Arabia	302 subjects 207 females 95 males	18–55 years	186 with prior GERD symptoms	*Coffee consumption increased risk of GERD* in 144 subjects (77.4% of the sample) No significant effect of sex and age
Arivan and Deepanjali, 2018 [43]	Cross-sectional survey using a validated symptom score	India	358 subjects 188 males 170 females	20.3 ± 1.5 (S.D.) years	193 without symptoms; 115 with at least a weekly episode of regurgitation: Diagnosis of GERD in 18 subjects	*GERD symptoms were more frequent in subjects frequently drinking tea and coffee*: OR = 4.65 (1.2–17.96); *p* = 0.026 GERD symptoms were not affected by gender or body mass index.
Mehta et al., 2019 [68]	Data collected from the Nurses’ Health Study, an ongoing prospective cohort study, started in 1989	USA	7961 women with GERD	42–62 years	Only females	*Coffee consumption and risk of GERD symptoms compared to no intake* *Total risk for any coffee* Intake ≥6 cups/day, HR = 1.34 (1.13–1.59) *Caffeinated coffee* <1 cup/day = HR = 1.11 (1.03–1.19) 1–3 cups/day = HR = 1.08 (1.03–1.14) 4–5 cups/day = 1.14 (1.02–1.27) ≥6 cups/day = 1.23 (1.00–1.50) *Decaffeinated coffee* <1 cup/day = HR = 1.05 (0.99–1.11) 1–3 cups/day = HR = 1.19 (1.10–1.28) 4–5 cups/day = 1.02 (0.73–1.40) ≥6 cups/day = 1.48 (0.92–2.39) *Replacing 2 servings of coffee/day by water* No risk: HR = 0.96 (0.92–1.00)
Correia et al., 2020 [35]	Qualitative intervention study	USA	51 subjects 45 females 6 males	29–83 years	All were patients with functional dyspepsia	*Reduction of the pre-operative median value (interquartile range) for reflux* from 4.00 (3.00) to 1.00 (1.00) *p* < 0.001 upon substitution of caffeinated or decaffeinated coffee by a non-caffeinated coffee substitute (roasted malt barley, roasted chicory, and roasted rye) for 1 month.
Green et al., 2020 [69]	Observational study, data from European participants in the UK Biobank	UK	379,713 subjects	Controls: 57.0 ± 8.0 years GERD cases: 59.3± 7.4 years	355,744 controls 345,744 males 33,969 GERD cases 15,823 males	*Limited association between coffee consumption and GERD* (241016 drinkers, 2.6 ± 2.1 cups/day in both groups): OR = 1.18 (0.88–1.58)
Single studies: Barrett’s esophageus						
Sajja et al., 2016 [70]	Cross-sectional study	USA	2038 veterans 310 BE cases 1728 without BE	60 ± 90.2 years for controls 61.6 ± 7.6 years for BE cases	1869 males (1567 controls and 302 BE cases) 169 females (161 controls and 8 BE cases)	*No association between coffee consumption and risk of BE* when population adjusted for confounders (including sex and race) Adjusted OR = 1.04 (0.76–1.42) for coffee drinkers compared to non-coffee drinkers
Filiberti et al., 2017 [59]	Retrospective case-control study: patients with Barrett’s esophageus (BE)	Italy	1420 subjects 766 males 654 females	53.7 ± 14.1 years for controls 56.2 ± 15.2 years for BE patients	619 controls (252 males, 367 females) 339 BE patients (229 males, 110 females)	*BE risk versus control:* *-higher in former coffee drinkers, irrespective of levels of exposure* ≤1 cup/day: OR = 3.76 (1.33–10.6) >1 cup/day: OR = 3.79 (1.31–11.0); test for linear trend (TLT) *p* = 0.006) -*higher with duration* >30 years: OR = 4.18 (1.43–12.3) -*higher for late quitters* ≤3 years after stopping: OR = 5.95 (2.19–16.2). -*higher in subjects who started drinking coffee at a later age > 18 years*: OR = 6.10 (2.15–17.3) -*no association in current drinkers*

Abbreviations: 95% CI: 95% confidence interval; HR: hazard ratio; OR: odds ratio; RR: relative risk; HRT: hormone replacement therapy.

**Table 2 nutrients-14-00399-t002:** Coffee and the risk of developing gallstones.

Authors	Study Design	Country	Size of the Population	Age (Years)	Number of Cases and Controls	Outcome
Reviews and meta-analyses						
Zhang et al., 2015 [89]	Meta-analysis of one case-control study and 5 prospective cohort studies	Europe (Italy, Sweden, UK) America	227,749 individuals		216,272 controls 11,477 cases	*Risk reduction of gallstone disease with coffee consumption*: - Case-control study: OR = 0.99 (95% CI = 0.64–1.53) - Prospective cohort studies: RR = 0.83 (95% CI = 0.76–0.89) - Increment of 1 cup/day: RR = 0.95 (95% CI = 0.91–1.00; *p* = 0.049). *Nonlinear association between coffee consumption and risk of gallstone disease, compared with no coffee consumption* - 2 cups/day, summary RR: 0.89 (95% CI = 0.79–0.99) - 3 cups/day, summary RR: 0.85 (95% CI = 0.76–0.94) - 4 cups/day, summary RR: 0.81 (95% CI = 0.72–0.92) - 5 cups/day, summary RR: 0.78 (95% CI = 0.67–0.90) - 6 cups/day, summary RR: 0.75 (95% CI = 0.64–0.88)
Kotrotsios et al., 2019 [90]	Review of epidemiological studies published between 1973 and 2018	Europe Asia				*Risk reduction of gallstone disease with coffee consumption*
Single studies						
La Vecchia et al., 1991 [91]	Case-control study	Italy	1317 participants 762 men and 555 women	<45–74 years	1122 controls (683 men and 436 women) and 195 cases (76 men and 119 women)	*No association between coffee consumption and gallstones* - 1 cup/day, RR: 1.1 (95% CI = 0.7–1.9) - 2 cups/day, RR: 0.9 (95% CI = 0.5–1.5) - 3 cups/day, RR: 1.0 (95% CI = 0.6–1.7) - ≥4 cups/day, RR: 0.9 (95% CI = 0.5–1.6)
Misciagna et al., 1996 [92]	Prospective cohort study	Italy	1962 participants, 1162 men and 800 women	30–69 years	1858 controls 104 cases (55 men and 49 women in the 7-year follow-up	*Risk reduction of gallstone disease with coffee consumption* Coffee consumers vs. non-consumers: OR: 0.75 (95% CI = 0.47–1.19)
Kratzer et al., 1997 [93]	Epidemiological study based on questionnaires	Germany	1116 participants, 656 men and 460 women	18–65 years	1049 controls (618 men and 431 women) and 67 cases (38 men and 29 women)	*No association between caffeine consumption and gallstones*
Sahi et al., 1998 [94]	Epidemiological study based on questionnaires	USA	16,787 men	unknown	15,786 controls 1019 cases	*No association between coffee consumption and gallstones* No measurement of the amount of coffee consumed
Leitzman et al., 1999 [95]	Prospective cohort study Health Professional Follow-up study (HPSF)	USA	46,008 men	40–75 years	44,927 controls 1081 cases	*Risk reduction of gallstone disease with coffee consumption* - 2–3 cups/day, adjusted RR: 0.60 (95% CI = 0.42–0.86) - ≥4 cups/day, adjusted RR: 0.55 (95% CI = 0.33–0.92) *Risk reduction of gallstone disease with caffeine consumption* - >800 vs. ≤25 mg/day, adjusted RR: 0.55 (95% CI = 0.35–0.87)
Ruhl & Everhart 2000 [96]	Cohort study Third National Health and Nutrition Examination Survey (NHANES III)	USA	13,983 participants 6675 men and 7263 women	20–74 years	1993 cases 578 men and 1415 women	*No association between coffee consumption and gallstones vs. no coffee consumption* *Total risk in men:* - 1–2 cups/day, adjusted PR: 1.07 (95% CI = 0.75–1.51) - >2 cups/day, adjusted PR: 0.89 (95% CI = 0.59–1.33) *Total risk in women:* - 1–2 cups/day, adjusted PR: 0.91 (95% CI = 0.72–1.14) - >2 cups/day, adjusted PR: 0.87 (95% CI = 0.70–1.08) *Decreased risk in women with previously diagnosed gallbladder disease:* - 1–2 cups/day, adjusted PR: 078 (95% CI = 0.59–1.03) - >2 cups/day, adjusted PR: 0.76 (95% CI = 0.56–1.02)
Leitzman et al., 2002 [97]	Prospective cohort study (Nurses’ Health Study, NHS)	USA	80,898 women	34–59 years et entry, 20 years follow-up	73,087 controls 7811 women with cholecystectomy	*Risk reduction of gallstone disease with coffee consumption* *Caffeinated coffee consumption compared to none:* - ≤1 cup/day, multivariate RR: 0.92 (95% CI = 0.87–0.98) - 2–3 cups/day, multivariate RR: 0.82 (95% CI = 0.78–0.87) - ≥4 cups/day, multivariate RR: 0.77 (95% CI = 0.71–0.83) *Decaffeinated coffee consumption compared to none:* - ≤1 cup/day, multivariate RR: 1.07 (95% CI = 1.02–1.13) - 2–3 cups/day, multivariate RR: 1.06 (95% CI = 0.98–1.14) - ≥4 cups/day, multivariate RR: 1.13 (95% CI = 1.00–1.28)
Ishizuka et al., 2003 [98]	Case-control study	Japan	7063 men	unknown	6887 controls 174 cases	*No association between coffee consumption and gallstones* *Coffee and prevalent gallstones (compared to no coffee consumption):* - 1–2 cups/day, adjusted OR: 1.0 (95% CI = 0.6–1.5) - 3–4 cups/day, adjusted OR: 0.9 (95% CI = 0.6–1.4) - ≥5 cups/day, adjusted OR: 1.4 (95% CI = 0.9–2.2) *Caffeine and prevalent gallstones (compared to consumption < 100 mg/day):* - 100–199 mg/day, adjusted OR: 1.1 (95% CI = 0.7–1.7) - 200–299 mg/day, adjusted OR: 0.8 (95% CI = 0.5–1.3) - ≥300 mg/day, adjusted OR: 1.4 (95% CI = 0.9–2.2) *Coffee and unknown gallstones (compared to no coffee consumption):* - 1–2 cups/day, adjusted OR: 0.9 (95% CI = 0.6–1.5) - 3–4 cups/day, adjusted OR: 0.7 (95% CI = 0.4–1.1) - ≥5 cups/day, adjusted OR: 1.0 (95% CI = 0.6–1.8) *Caffeine and unknown gallstones (compared to consumption < 100 mg/day):* - 100–199 mg/day, adjusted OR: 1.0 (95% CI = 0.6–1.6) - 200–299 mg/day, adjusted OR: 0.8 (95% CI = 0.5–1.3) - ≥300 mg/day, adjusted OR: 0.8 (95% CI = 0.5–1.5)
Walcher et al., 2010 [99]	Epidemiological study based on questionnaires	Germany	2147 participants 1036 men and 1111 women	18–65 years	1976 controls 171 cases	*No association between coffee/caffeine consumption and gallstones* - Consumption vs. none, OR: 0.77 (95% CI = 0.42–1.42) - Several times/week vs. never, OR: 0.67 (95% CI = 0.46–0.99) - Daily vs. never, OR: 0.70 (95% CI = 0.40–1.22)
Nordenvall et al., 2014 [100]	Cohort study Swedish Mammography Cohort and Cohort of Swedish Men	Sweden	71,925 participants 40,936 men and 30898 women	Born 1914–1948	69,906 controls, 2019 cases, 962 men and 1057 women	*Inverse association between coffee consumption and gallstones in premenopausal women or HRT users but not in other women or men* *Coffee and gallstones in all men vs. coffee consumption < 2 cups/day:* - 2–3 cups/day, multivariate-adjusted HR: 1.04 (95% CI = 0.86–1.25) - 4–5 cups/day, multivariate-adjusted HR: 0.96 (95% CI = 0.78–1.18) - ≥6 cups/day, multivariate-adjusted HR: 0.96 (95% CI = 0.75–1.24) *Coffee and gallstones in all women vs. coffee consumption < 2 cups/day:* - 2–3 cups/day, multivariate-adjusted HR: 0.88 (95% CI = 0.74–1.03) - 4–5 cups/day, multivariate-adjusted HR: 0.67 (95% CI = 0.55–0.82) - ≥6 cups/day, multivariate-adjusted HR: 0.58 (95% CI = 0.44–0.78) *Coffee and gallstones in groups of women (stratified by menopausal status) vs. coffee consumption < 2 cups/day:* *Premenopausal* - 2–3 cups/day, multivariate-adjusted HR: 0.64 (95% CI = 0.41–1.01) - 4–5 cups/day, multivariate-adjusted HR: 0.37 (95% CI = 0.21–0.67) - ≥6 cups/day, multivariate-adjusted HR: 0.17 (95% CI = 0.05–0.55) *Postmenopausal, current user of HRT* - 2–3 cups/day, multivariate-adjusted HR: 0.77 (95% CI = 0.62–0.97) - 4–5 cups/day, multivariate-adjusted HR: 0.59 (95% CI = 0.45–0.78) - ≥6 cups/day, multivariate-adjusted HR: 0.44 (95% CI = 0.28–0.70) *Postmenopausal, past user of HRT* - 2–3 cups/day, multivariate-adjusted HR: 1.17 (95% CI = 0.69–1.95) - 4–5 cups/day, multivariate-adjusted HR: 0.75 (95% CI = 0.40–1.39) - ≥6 cups/day, multivariate-adjusted HR: 0.77 (95% CI = 0.33–1.76) *Postmenopausal, never user of HRT* - 2–3 cups/day, multivariate-adjusted HR: 1.09 (95% CI = 0.78–1.53) - 4–5 cups/day, multivariate-adjusted HR: 0.96 (95% CI = 0.66–1.39) - ≥6 cups/day, multivariate-adjusted HR: 1.09 (95% CI = 0.68–1.74)
Nordestgaard et al., 2019 [101]	Prospective observational study	Denmark	101,190 individuals, 47,001 men and 54,189 women	58 (48–67) years	98,957 controls and 2233 cases, 8 years follow-up	*Risk reduction of gallstone disease with coffee consumption* Compared to no coffee consumption 0.1–3 cups/day: HR = 0.86 (0.75–0.99) 3.1–6 cups/day: HR = 0.80 (0.69–0.93) >3 cups/day: HR = 0.83 (0.66–1.03)

Abbreviations: 95% CI: 95% confidence interval; HR: hazard ratio; OR: odds ratio; RR: relative risk; HRT: hormone replacement therapy.

**Table 3 nutrients-14-00399-t003:** Effects of coffee on recovery after abdominal surgery.

Authors	Study Design	Type of Surgery	Size of the Population	Outcome
Eamudomkarn et al., 2018 [113]	Systematic review and meta-analysis of 6 randomized control studies	3 studies on cesarean deliveries 2 on colorectal cancers 1 on gynecologic cancer surgery	601 cases	*Reference: water or no intervention* *Time to first flatus:* Decreased time to first flatus (MD, −7.14 h; 95% CI, −10.96 to −3.33 h). *Time to first bowel sound* (434 participants undergoing cesarean delivery or gynecologic cancer surgery): Shorter time to first audible bowel sound (MD, −4.17 h; 95% CI, −7.88 to −0.47 h). *Time to first defecation:* Reduced time to first defecation (MD, −9.98 h; 95% CI, −16.97 to −2.99 h) *Time to tolerance of solid food* (476 participants) Shorter time to tolerance of solid food (MD, −15.55 h; 95% CI, −22.83 to −8.27 h). *Postoperative nausea* (359 participants undergoing cesarean delivery and gynecologic cancer surgery) No significant difference in the risk of postoperative nausea (RR, 0.61; 95% CI, 0.27–1.36). *Length of hospital stay* (476 participants) Shorter length of hospital stay (MD, −0.74 days; 95% CI, −1.14 to −0.33 days),
Cornwall et al., 2020 [114]	Systematic review and meta-analysis of 7 randomized control studies	150 cesarean deliveries 114 gynecologic resections 342 colorectal resections	606 cases: 317 patients and 289 controls	*Reference: water or no intervention* *Time to first flatus:* No significant effect of coffee on time to first flatus (MD = −3.6 h, 95% CI: 0.8, −7.96 h, *p* = 0.11). *Time to first bowel sound* (264 participants) Reduced time to first bowel sounds or sensation of bowel movement (MD = −3.3 h, 95% CI: −0.6, −6.0 h, *p* = 0.02). *Time to first defecation:* Reduced time to first defecation (MD = −11.8 h, 95% CI: −2.2, −18.5 h, *p* < 0.00001) *Time to tolerance of solid food* (280 participants) Reduced time to tolerance of solid food (MD = −17.1 h, 95% CI: −2.9 to −31.2 h, *p* = 0.02). *Postoperative nausea*: assessed in 359 patients undergoing cesarean delivery and gynecologic cancer surgery. No significant difference in the risk of postoperative nausea (RR, 0.61; 95% CI, 0.27–1.36). *Length of hospital stay* (556 participants) No significant association between coffee consumption and length of hospital stay (MD = −1.9 days, 95% CI: 1.7, −5.4 days, *p* = 0.30)
Gkegkes et al., 2020 [115]	Systematic review and meta-analysis of 4 randomized control studies	3 studies on colorectal surgery 1 on gynecological interventions	341 patients, 156 cases and 185 controls	*Reference: water or no intervention* *Time to first flatus:* Reduced time to first flatus (MD = −10.02 h; 95% CI −15.54 to –4.50 h) *Time to first bowel movement* (264 participants) Reduced time to first bowel sensation of bowel movement (MD = −12.09 h; 95% CI: −15.26 to –8.92 h) *Time to first defecation:* Reduced time to first defecation (MD = −16.14 h; 95% CI: −18.59 to 13.70 h) *Time to tolerance of solid food:* Reduced time to tolerance of solid food (MD = −1.31 h, 95% CI: −1.83 to −0.79 h) *Length of hospital stay:* No significant association between coffee consumption and length of hospital stay (MD = −3.18 days; 95% CI: −8.25 to 1.89 days)
Kane et al., 2020 [116]	Systematic review and meta-analysis of 4 randomized control studies published since 2012	3 studies on resection of colon/rectum 3 studies on gynecological interventions		*Reference: water or no intervention* *Time to first flatus* (403 patients): Reduced time to first flatus (MD = −6.96 h; 95% CI: −9.53 to –4.38 h) *Time to first defecation* (231 patients): Reduced time to first defecation (MD = −9.38 h; 95% CI: −17.60 to 1.16 h) *Time to tolerance of solid food* (253 patients): Reduced time to tolerance of solid food (MD = −9.52 h, 95% CI: −18.19 to −0.85 h) *Length of hospital stay* (311 patients): Reduced length of hospital stay (MD = −2.81 days; 95% CI: −7.14 to 1.51 days)
Watanabe et al., 2021 [117]	Systematic review and meta-analysis of 13 randomized control trials published since 2012 9 ongoing trials were included	6 trials on colorectal surgery 5 trials on cesarean section 2 trials on gynecological surgery	1246 patients	*Reference: water or no intervention* *Time to first flatus:* Reduced time to first flatus (MD = −4.3 h, 95% CI: −8.5 to −0.07 h, *p* = 0.11). *Time to first bowel sound:* Reduced time to first bowel sounds (MD = −4.3 h, 95% CI: −7.1 to −1.5 h). *Time to first defecation:* Reduced time to first defecation (MD =−10 h, 95% CI = −14 to −5.6 h) *Time to tolerance of solid food:* Reduced time to tolerance of solid food (MD = −9.9 h, 95% CI: −14 to −5.9 h). *Length of hospital stay* (556 participants) No significant association between coffee consumption and length of hospital stay (MD = −1.5 days, 95% CI: −2.7 to −0.3 days)

Abbreviations: MD: mean difference; 95% CI: 95% confidence interval; RR: relative risk.

## Data Availability

Not applicable.

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
