# Peer review of "Effects of Coffee on the Gastro-Intestinal Tract: A Narrative Review and Literature Update"

_nutrients, 2022, doi:10.3390/nu14020399_

Round 1
Reviewer 1 Report
The manuscript under appreciation is a narrative review and literature update of the effects of coffee on the gastro-intestinal tract, as clear and concisely is stated in the title. In fact, it is expected much more from a scientific review.
Coffee beverages are very complex matrices, composed by a large range of compounds, modulated by the roasting conditions and mode of preparation, and that can have different and sometimes contradictory effects on gastro-intestinal tract. Even if, as stated “This review is not intended at considering the detailed mechanisms of action on coffee on these processes since these aspects have been developed in several detailed and recent reviews”, a relationship about which coffee components should be involved in the different topic covered related with the gastro-intestinal tract should be established. This could be the key to have a rationale that could explain why there are contradictory conclusions obtained from the studies listed. Indeed, the paragraphs listing the different studies should be integrated in order to (re)interpret the data available in light of the knowledge of coffee composition and intake in its different forms by individuals.
Other specific points that need attention are the following:
Line 19. Keywords have not been provided.
Line 32. References are needed.
Line 45. As emphasis is given on coffee temperature, it would be useful to explain the effect of temperature on salivary alpha-amylase activity.
Line 74. It is not clear when the sample is coffee or only an isolated compound used as standard, such as caffeine.
Lines 88-91. Why this regional difference? Consumers? Samples? Methodology? Other?
Lines 91-93. The information about the temperature and coffee composition used in the experiments should be provided. Was the concentration of caffeine the same between samples? In fact, caffeine is not affected by the roasting of coffee.
Lines 102-107. Polysaccharides are reported to have a gastro-protective effect. Can coffee polysaccharides have this effect?
Line 121. Use Celsius degrees, not Fahrenheit.
Line 128-129. Vague sentence. Please explain the differences in green beans processing.
Line 138. It would be useful an explanation about the meaning of the “Odds Ratio”.
Table 1. Ref. 42 has association of coffee with gastro-esophageal reflux (GERD) although listed in the no association set.
Table 1. It is necessary to explain why some studies concluded for reduced risk of GERD in coffee consumers and many others concluded that an increased risk occurs. Lines 151-218.
Line 169. Is the caffeine the active principle?
Line 191. What have the listed foods in common?
Line 193. Please explain what “Barrett's esophagus” is.
Line 206. Please explain what “other lifestyle habits” are.
Line 208. The sentence is very complicated with a double negation: “low level coffee consumption does not seem to not strongly influence the occurrence of GERD while higher levels of consumption might somewhat increase the risk”.
Lines 251-254. Arabinogalactans and other coffee polysaccharides may also exert this effect, not only chlorogenic acids.
Line 261. Which coffee components?
Line 264. Can coffee bile acids sequestration influence secretion effectiveness?
Table 2. It should be useful to check if the parameter that was considered relevant in some studies have been taken into consideration in those where no relation has been noticed.
Lines 330-333. Can other coffee components beyond caffeine, such as chlorogenic acids, hardly separated from caffeine, have similar effects?
Lines 351-352. Please indicate which coffee compounds can stimulate colon motility.
Line 365. This chapter should be more than a range of paragraphs describing different studies. Connectivity among them should be searched.
Lines 439-440. Please explain why.
Line 441. Too descriptive chapter lacking integration of concepts. Coffee contains pro-inflammatory and anti-inflammatory compounds that can provide contradictory results.
Lines 472 and 506. Uniformize the concept of “dietary fibers” and “polysaccharides”.
Lines 506 and 513. Uniformize the concept of “polyphenols” and “chlorogenic acids”.
Lines 536-538. Why have spent coffee grounds this effect?
Line 552. What are the main ideas for the proposed studies?
Lines 580-583. Are correct the temperatures reported?
Line 592. The volume of coffee cups varies according to the mode of preparation. What is the volume of a “cup”?
Line 597. Which type of coffee?
Line 601. Does his study refer to American coffee, in large cups?
Line 618. Which type of studies are required?
Line 737-751. As conclusions seems to be the same as a previous published review, what is its relevance?
Lines 752-759. Delete this paragraph.
In conclusion, it is my opinion that considering the above-mentioned comments/suggestions the manuscript needs corrections.
Author Response
Please see attachment author-coverletter-16541617.v2.doxc for Table 3 and point-to-point response to reviewer 1 in the author's notes files under report notes. Direct loading of both files under their initial name did not work.
Below I also add a copy of Table 3 just to prevent that its insertion in the final draft would be forgotten.
Table 3: Effects of coffee on recovery after abdominal surgery
|
Authors |
Study design |
Type of surgery |
Size of the population |
Outcome |
|
Eamudomkarn et al. 2018 [115] |
Systematic review and meta-analysis of 6 randomized control studies
|
3 studies on cesarean deliveries 2 on colorectal cancers 1 on gynecologic cancer surgery |
601 cases |
Reference: water or no intervention Time to first flatus: Decreased time to first flatus (MD, −7.14 h; 95% CI, −10.96 to −3.33 h). Time to first bowel sound (434 participants undergoing cesarean delivery or gynecologic cancer surgery): Shorter time to first audible bowel sound (MD, −4.17 h; 95% CI, −7.88 to −0.47 h). Time to first defecation: Reduced time to first defecation (MD, −9.98 h; 95% CI, −16.97 to −2.99 h) Time to tolerance of solid food (476 participants) Shorter time to tolerance of solid food (MD, −15.55 h; 95% CI, −22.83 to −8.27 h). Postoperative nausea (359 participants undergoing cesarean delivery and gynecologic cancer surgery) No significant difference in the risk of postoperative nausea (RR, 0.61; 95% CI, 0.27-1.36). Length of hospital stay (476 participants) Shorter length of hospital stay (MD, −0.74 days; 95% CI, −1.14 to −0.33 days), |
|
Cornwall et al. 2020 [116] |
Systematic review and meta-analysis of 7 randomized control studies
|
150 cesarean deliveries 114 gynecologic resections 342 colorectal resections
|
606 cases: 317 patients and 289 controls |
Reference: water or no intervention Time to first flatus: No significant effect of coffee on time to first flatus (MD = -3.6 h, 95% CI: 0.8, -7.96 h, P = 0.11). Time to first bowel sound (264 participants) Reduced time to first bowel sounds or sensation of bowel movement (MD = -3.3 h, 95% CI: -0.6, -6.0 h, P = 0.02). Time to first defecation: Reduced time to first defecation (MD = -11.8 h, 95% CI: -2.2, -18.5 h, p < 0.00001) Time to tolerance of solid food (280 participants) Reduced time to tolerance of solid food (MD = -17.1 h, 95% CI: -2.9 to -31.2 h, P = 0.02). Postoperative nausea: assessed in 359 patients undergoing cesarean delivery and gynecologic cancer surgery. No significant difference in the risk of postoperative nausea (RR, 0.61; 95% CI, 0.27-1.36). Length of hospital stay (556 participants) No significant association between coffee consumption and length of hospital stay (MD = -1.9 days, 95% CI: 1.7, -5.4 days, P = 0.30) |
|
Gkegkes et al. 2020 [117] |
Systematic review and meta-analysis of 4 randomized control studies
|
3 studies colorectal surgery 1 on gynecological interventions |
341 patients, 156 cases and 185 controls |
Reference: water or no intervention Time to first flatus: Reduced time to first flatus (MD = –10.02 h; 95% CI –15.54 to –4.50 h) Time to first bowel movement (264 participants) Reduced time to first bowel sensation of bowel movement (MD = –12.09 h; 95% CI –15.26 to –8.92 h) Time to first defecation: Reduced time to first defecation (MD = –16.14 h; 95% CI –18.59 to 13.70 h) Time to tolerance of solid food: Reduced time to tolerance of solid food (MD = -1.31 h, 95% CI: -1.83 to -0.79 h) Length of hospital stay: No significant association between coffee consumption and length of hospital stay (MD = –3.18 days; 95% CI –8.25 to 1.89 days) |
|
Kane et al. 2020 [118] |
Systematic review and meta-analysis of 4 randomized control studies published since 2012
|
3 studies on resection of colon/rectum 3 studies on gynecological interventions |
|
Reference: water or no intervention Time to first flatus (403 patients): Reduced time to first flatus (MD = –6.96 h; 95% CI –9.53 to –4.38 h) Time to first defecation (231 patients): Reduced time to first defecation (MD = –9.38 h; 95% CI –17.60 to 1.16 h) Time to tolerance of solid food (253 patients): Reduced time to tolerance of solid food (MD = -9.52 h, 95% CI: -18.19 to -0.85 h) Length of hospital stay (311 patients): Reduced length of hospital stay (MD = –2.81 days; 95% CI –7.14 to 1.51 days) |
|
Watanabe et al. 2021 [119] |
Systematic review and meta-analysis of 13 randomized control trials published since 2012 9 ongoing trials were included |
6 trials on colorectal surgery 5 trials on cesarean section 2 trials on gynecological surgery |
1246 patients |
Reference: water or no intervention Time to first flatus: Reduced time to first flatus (MD = -4.3 h, 95% CI: -8.5 to -0.07 h, P = 0.11). Time to first bowel sound: Reduced time to first bowel sounds (MD = -4.3 h, 95% CI: -7.1 to -1.5 h). Time to first defecation: Reduced time to first defecation (MD =−10 h, 95% CI = −14 to −5.6 h) Time to tolerance of solid food: Reduced time to tolerance of solid food (MD = -9.9 h, 95% CI: -14 to -5.9 h). Length of hospital stay (556 participants) No significant association between coffee consumption and length of hospital stay (MD = -1.5 days, 95% CI: -2.7 to -0.3 days) |
Abbreviations: MD: mean difference; 95% CI: 95% Confidence Interval; RR: Relative risk.

Reviewer 2 Report
This is an interesting and a well-organized review article.
However, an error is found in Table 1. The outcome of the manuscript by Bhatia et al. 2011 is thought to be 'no association between coffee consumption and GERD'. Therefore, you would add the word 'No'.
Author Response
Reply to reviewer's comments
This is an interesting and a well-organized review article.
Thanks for the reviewer's very positive and appreciative comment.
However, an error is found in Table 1. The outcome of the manuscript by Bhatia et al. 2011 is thought to be 'no association between coffee consumption and GERD'. Therefore, you would add the word 'No'.
Thanks for picking up this critical mistake which escaped my attention and that of all my colleagues who read the final version of the manuscript. I am really happy that you noticed this. Thanks for your careful reviewing.
Round 2
Reviewer 1 Report
Although the author refer to have point-to-point responses to Reviewer 1 in the author's notes files, it fact no responses have been provided except the inclusion of a new Table (Table 3), both in a Word document (author_response.docx) and in Author's Notes.
To accurately analyse the revised version of manuscript, and avoid a new extensive read of the manuscript, it is necessary to have this complementary text. with the author's point of view of the comments/suggestions made.
Author Response
Please find below my detailed answers to reviewer 1 comments:
NUTRIENTS-1530328: REPLY TO REVIEWER 1.
The manuscript under appreciation is a narrative review and literature update of the effects of coffee on the gastro-intestinal tract, as clear and concisely is stated in the title. In fact, it is expected much more from a scientific review.
Coffee beverages are very complex matrices, composed by a large range of compounds, modulated by the roasting conditions and mode of preparation, and that can have different and sometimes contradictory effects on gastro-intestinal tract. Even if, as stated “This review is not intended at considering the detailed mechanisms of action on coffee on these processes since these aspects have been developed in several detailed and recent reviews”, a relationship about which coffee components should be involved in the different topic covered related with the gastro-intestinal tract should be established. This could be the key to have a rationale that could explain why there are contradictory conclusions obtained from the studies listed. Indeed, the paragraphs listing the different studies should be integrated in order to (re)interpret the data available in light of the knowledge of coffee composition and intake in its different forms by individuals.
Whenever possible, the role of the different coffee components was added to the various effects of coffee on the organs and functions of the digestive tract
Other specific points that need attention are the following:
Line 19. Keywords have not been provided.
Keywords have now been provided, see lines 62, 63.
Line 32. References are needed.
References have been provided, see page 1 line 78.
Line 45. As emphasis is given on coffee temperature, it would be useful to explain the effect of temperature on salivary alpha-amylase activity.
A sentence was added on this point, see page 2 lines 54-56.
Line 74. It is not clear when the sample is coffee or only an isolated compound used as standard, such as caffeine.
A comment was added on this point, the only available data concern caffeine, see page 2 lines 87-88.
Lines 88-91. Why this regional difference? Consumers? Samples? Methodology? Other?
There is no explanation for this and I do not have one, text on lines 104-107.
Lines 91-93. The information about the temperature and coffee composition used in the experiments should be provided. Was the concentration of caffeine the same between samples? In fact, caffeine is not affected by the roasting of coffee.
The authors do not mention the temperature used but only specify that it is conventional convection roasting, so I took out any notion of temperature, see lines 109-110.
Lines 102-107. Polysaccharides are reported to have a gastro-protective effect. Can coffee polysaccharides have this effect?
I have no answer to this question and did not find any literature data, see lines 124-129.
Line 121. Use Celsius degrees, not Fahrenheit.
Done, see page 2 lines 54-56.
Line 128-129. Vague sentence. Please explain the differences in green beans processing.
I could not find any further clarification on this point, there are no other literature data, see lines 150-154.
Line 138. It would be useful an explanation about the meaning of the “Odds Ratio”.
Done, see lines 163-165.
Table 1. Ref. 42 has association of coffee with gastro-esophageal reflux (GERD) although listed in the no association set.
This is an error remaining from my side, it has now been corrected in Table 1.
Table 1. It is necessary to explain why some studies concluded for reduced risk of GERD in coffee consumers and many others concluded that an increased risk occurs. Lines 151-218.
I added some information of possible explanation but there is no straight explanation to this because of the variability of the study conditions, type of countries, habits of consumption, types of coffee, processing, preparations and others that have been summarized more relevant to this specific effect, see lines 261-276.
Line 169. Is the caffeine the active principle?
I added a sentence of clarification on lines 199-201.
Line 191. What have the listed foods in common?
The reason for the choice of the different foods is now given, see lines 221-226.
Line 193. Please explain what “Barrett's esophagus” is.
Done, see lines 228-229.
Line 206. Please explain what “other lifestyle habits” are.
It is not explained by the authors and I do not want to speculate on their hypothesis but it is widely recognized that non-coffee drinkers are a population different from coffee drinkers. I suppressed the sentence on this, see lines 240-242.
Line 208. The sentence is very complicated with a double negation: “low level coffee consumption does not seem to not strongly influence the occurrence of GERD while higher levels of consumption might somewhat increase the risk”.
Sorry for this awkward sentence. The second negation was an error, it is now deleted, see line 243.
Lines 251-254. Arabinogalactans and other coffee polysaccharides may also exert this effect, not only chlorogenic acids.
I added a sentence on polysaccharides but there is no direct evidence from the literature, see lines 317-319.
Line 261. Which coffee components?
I mentioned other components such as chlorogenic acids and polysaccharides, see lines 324-325.
Line 264. Can coffee bile acids sequestration influence secretion effectiveness?
I did not find a clear answer to this question.
Table 2. It should be useful to check if the parameter that was considered relevant in some studies have been taken into consideration in those where no relation has been noticed.
The question is too vague, what parameter, I do not understand the question?
Lines 330-333. Can other coffee components beyond caffeine, such as chlorogenic acids, hardly separated from caffeine, have similar effects?
I did not find any data on this.
Lines 351-352. Please indicate which coffee compounds can stimulate colon motility.
The role of coffee components other than caffeine has not been studied. I mentioned the possible role of indirect factors, see lines 429-434.
Line 365. This chapter should be more than a range of paragraphs describing different studies. Connectivity among them should be searched.
This has been corrected. I have built a table (table 3) summarizing all the data for the 5 meta-analyses which allows to be briefer and more synthetic in the commentary of the data, see table joined and comments on lines 440-487 and 552-557.
Lines 439-440. Please explain why.
This is now part of the new more integrated paragraph on lines 440-487.
Line 441. Too descriptive chapter lacking integration of concepts. Coffee contains pro-inflammatory and anti-inflammatory compounds that can provide contradictory results.
This is now part of the new more integrated paragraph on lines 440-487.
Lines 472 and 506. Uniformize the concept of “dietary fibers” and “polysaccharides”.
This has been clarified, see line 596.
Lines 506 and 513. Uniformize the concept of “polyphenols” and “chlorogenic acids”.
Done, see line 636.
Lines 536-538. Why have spent coffee grounds this effect?
I did not find any explanation for this.
Line 552. What are the main ideas for the proposed studies?
Relevant comments were added, see lines 678-681.
Lines 580-583. Are correct the temperatures reported?
These temperatures are exactly those that the authors indicate in their paper.
Line 592. The volume of coffee cups varies according to the mode of preparation. What is the volume of a “cup”?
In the whole coffee literature, a cup of coffee is a standard of 150 mL, equivalent to an American mug and there is no need to indicate this, everybody is in agreement.
Line 597. Which type of coffee?
American type of coffee.
Line 601. Does his study refer to American coffee, in large cups?
Yes, it does.
Line 618. Which type of studies are required?
The field on coffee and pancreas cancer has been quite polemical for 50 years and it would be much too long to explain this here in particular given that smoking remains a confounding factor in this cancer type. This would go beyond the scope of this review, especially given that is was only intending to give a brief summary of what is known on the relationship between coffee consumption and cancer.
Line 737-751. As conclusions seems to be the same as a previous published review, what is its relevance?
The conclusion has been expressed differently since clearly my message did not come through and hence I dedided to suppress the reference in the conclusion. The present review is not just simply a repetition of the paper that was cited in the first version which concerns the gastro-intestinal mucosa and not the digestive function but there are some common conclusions.
Lines 752-759. Delete this paragraph.
Done.